# Molecular and Biochemical Mechanisms of Cardiomyopathy Development Following Prenatal Hypoxia—Focus on the NO System

**DOI:** 10.3390/antiox14060743

**Published:** 2025-06-16

**Authors:** Olena Popazova, Igor Belenichev, Nina Bukhtiyarova, Victor Ryzhenko, Nadia Gorchakova, Valentyn Oksenych, Oleksandr Kamyshnyi

**Affiliations:** 1Department of Histology, Cytology and Embryology, Zaporizhzhia State Medical and Pharmaceutical University, 69000 Zaporizhzhia, Ukraine; 2Department of Pharmacology and Medical Formulation with Course of Normal Physiology, Zaporizhzhia State Medical and Pharmaceutical University, 69000 Zaporizhzhia, Ukraine; belenichev.i.f@zsmu.edu.ua; 3Department of Clinical Laboratory Diagnostics, Zaporizhzhia State Medical and Pharmaceutical University, 69000 Zaporizhzhia, Ukraine; 4Department of Medical and Pharmaceutical Informatics and Advanced Technologies, Zaporizhzhia State Medical University, 69000 Zaporizhzhia, Ukraine; 5Department of Pharmacology, Bogomolets National Medical University, 01601 Kyiv, Ukraine; 6Faculty of Medicine, University of Bergen, 5020 Bergen, Norway; 7Department of Microbiology, Virology and Immunology, I. Horbachevsky Ternopil State Medical University, 46001 Ternopil, Ukraine; kamyshnyi_om@tdmu.edu.ua

**Keywords:** prenatal hypoxia, cardiomyopathy, endothelial dysfunction, mitochondrial dysfunction, oxidative stress, nitrosative stress, cardiomyocyte apoptosis, NO modulators, HSP70

## Abstract

Prenatal hypoxia (PH) adversely affects the development of the fetal heart, contributing to persistent cardiovascular impairments in postnatal life. A key component in regulating cardiac physiology is the nitric oxide (NO) system, which influences vascular tone, myocardial contractility, and endothelial integrity during development. Exposure to PH disrupts NO-related signaling pathways, leading to endothelial dysfunction, mitochondrial damage, and an escalation of oxidative stress—all of which exacerbate cardiac injury and trigger cardiomyocyte apoptosis. The excessive generation of reactive nitrogen species drives nitrosative stress, thereby intensifying inflammatory processes and cellular injury. In addition, the interplay between NO and hypoxia-inducible factor (HIF) shapes adaptive responses to PH. NO also modulates the synthesis of heat shock protein 70 (HSP70), a critical factor in cellular defense against stress. This review emphasizes the involvement of NO in cardiovascular injury caused by PH and examines the cardioprotective potential of NO modulators—Angiolin, Thiotriazoline, Mildronate, and L-arginine—as prospective therapeutic agents. These agents reduce oxidative stress, enhance endothelial performance, and alleviate the detrimental effects of PH on the heart, offering potential new strategies to prevent cardiovascular disorders in offspring subjected to prenatal hypoxia.

## 1. Introduction

Introduction. Post-hypoxic disorders of the cardiovascular system are among the leading causes of morbidity in newborns, occurring in 40–70% of children who have experienced prenatal hypoxia, according to various sources. These disorders are the starting point for many often serious diseases in both children and adults [1,2,3,4,5,6]. To this day, the mechanisms of post-hypoxic cardiac disturbances remain poorly understood, making it a relevant issue in pediatric cardiology [7,8]. The clinical presentation of this pathology during the acute phase is highly variable and can mimic other diseases, necessitating careful differential diagnosis from congenital heart defects, congenital myocarditis, and various cardiomyopathies [9]. To this day, there is no consensus on the step-by-step comprehensive therapy for post-hypoxic cardiac disorders [10]. Therefore, identifying new structural, molecular, and biochemical features of post-hypoxic cardiovascular disorders in newborns and developing pharmacotherapy strategies based on these findings is of scientific interest. According to current understanding, endothelial dysfunction and the associated disturbances in the NO system are fundamental to the development of many cardiovascular diseases [11,12]. Under the influence of hypoxia, infection, and other damaging factors, the functioning of the nitric oxide system is disrupted, leading to the development of pathology in various organs and systems, including the cardiovascular system [13,14]. However, the literature concerning the role of the NO system in the development of cardiovascular pathology in newborns and the potential cardioprotective effects of its modulators is quite limited. Several investigations have demonstrated that certain pharmacological agents exhibit cardio- and endothelium-protective effects by promoting NO synthesis and improving its bioavailability [15,16,17,18,19].

## 2. **Prenatal Hypoxia and Its Impact on Cardiovascular Development**

### 2.1. Prenatal Hypoxia and Its Consequences

Hypoxic changes in the myocardial energy metabolism lead to a rapid decrease in its contractile function (Figure 1). This is facilitated by certain anatomical and physiological characteristics of newborns, such as the diffuse type of coronary arteries with numerous anastomoses between the right and left coronary arteries, their small diameter, as well as the predominance of sympathetic nervous system influence, the tone of which is maintained by the preceding hypoxic condition of the central nervous system (CNS), known as the cerebrocardiac syndrome [20,21,22].

Fetal hypoxia disrupts autonomic regulation of coronary vasculature, compromises energy metabolism, and significantly reduces the production of high-energy (macroergic) compounds in cardiomyocyte mitochondria [23,24,25,26]. Acidosis, hypercatecholaminemia, hypoglycemia, and the deterioration of the blood’s rheological properties are key factors in the pathogenesis of hypoxic damage to the conduction cardiomyocytes in newborns and are the cause of various types of arrhythmias [27,28]. A well-known role in the development of post-hypoxic cardiac rhythm disturbances is played by disruptions in vegetative regulation [29,30]. The connection between hypoxic myocardial damage and various disturbances in cardiac rhythm and conductivity is evidenced by morphological and ultrastructural studies [31,32].

In the conduction system of the heart after prenatal hypoxia, signs of apoptosis and dystrophy are observed, with a certain correlation between the severity of morphological changes and clinically recognized cadence and conductivity unsettling influences [33,34]. The ultimate morphological result of hypoxic myocardial harm can be central dystrophy, which has two conceivable results: either total determination and reclamation of work, or the arrangement of central cardiosclerosis [35,36,37]. Currently, cardiomyopathies are understood as diseases of unclear etiology, primarily affecting the myocardium, where the contractile proteins of the heart muscle lose some of their properties, resulting in an insufficiently effective contraction of the heart muscle [38,39,40,41].

This, in turn, negatively impacts the entire circulatory system of the child—symptoms of heart failure arise and gradually worsen, accompanied by blood shunting from one circulatory circuit to another in the absence of morphological signs of active inflammation [37,42]. Cardiomyopathies are classified into primary (idiopathic) and secondary types [43]. Transient post-hypoxic myocardial ischemia is classified as a secondary cardiomyopathy and is predominantly observed in the first hours and days of a newborn’s life. Among the hemodynamic factors, transient pulmonary hypertension, increased blood pressure, and the closure of fetal communications play a significant role in the development of post-hypoxic cardiomyopathy, as they create an additional workload on the myocardium with reduced functional capacity [44,45,46].

Since the time of birth, and depending on its outcome, the level and degree of cardiovascular system damage will vary: neonatal pulmonary hypertension, persistence of fetal communications, myocardial dysfunction with chamber dilation, myocardial ischemia, and disturbances in heart rhythm and conductivity [47,48,49,50,51,52,53]. Hypoxia increases the workload on the heart, as the newborn experiences vasoconstriction in both the pulmonary and systemic circulations, resulting from catecholamine release and the direct effects of elevated carbon dioxide levels [23,26,54,55]. Blood return to the heart increases, raising the pressure in the right ventricle, which may become equal to the systemic arterial pressure. Myocardial blood flow is unable to fully supply the cardiomyocytes with oxygen, and consequently, the demand for oxygen rises. This leads to the development of coronary insufficiency and myocardial ischemia [37,56,57,58,59].

In children who have undergone both chronic intrauterine and perinatal hypoxia, a cardiovascular system maladaptation syndrome is observed during the neonatal period, accompanied by a prolonged (up to several months of life) increase in the activity of cardiac-specific enzymes [60,61]. This period can be considered transitional in terms of myocardial metabolism in newborns affected by hypoxia [62]. In children who underwent chronic intrauterine hypoxia, the cardiovascular maladaptation syndrome is of a transient, benign nature, with rapid reversal of clinical symptoms and almost complete absence of residual phenomena [63,64]. One in three children who experienced perinatal hypoxia have residual symptoms, such as minimal signs of pulmonary hypertension [65]. Valve insufficiency and reduced contractile ability of the ventricular myocardium are identified much less frequently. This dictates the need for prolonged outpatient monitoring of this group of children and appropriate medical interventions [64,65,66].

Identifying the type of autonomic reactivity helps determine the leading direction for corrective measures, preventing the formation of functional heart pathology in these patients in the future [67]. Prenatal hypoxia leads to an increased workload on the heart, as vasoconstriction of the vessels in both the systemic and pulmonary circulations occurs in the child due to disruptions in the nitric oxide system, nitrosative stress, and endothelial dysfunction [23,68,69,70,71]. All of this leads to the circulatory system failing to perform its function of providing oxygen to the working heart, resulting in myocardial ischemia.

On average, 30% of children who have experienced intrauterine hypoxia retain residual phenomena, such as minimal signs of pulmonary hypertension, reduced heart pump function, and disturbances in the autonomic regulation of cardiac activity [36,72]. Prenatal hypoxia negatively impacts the morphological and functional characteristics of the cardiovascular system at all stages of ontogenesis and may lead to the formation of a disproportionate development pattern of the heart, as well as morphological and functional disturbances in the conduction system [6,8].

The functional changes in the cardiovascular system seen in the post-hypoxic maladaptation syndrome are based on impaired neurohumoral regulation of vascular tone, transient neonatal pulmonary hypertension, drawn-out persistence of fetal communications (PFC), and a delay within the arrangement of the developing sort of cardiomyocyte digestion system [73,74,75]. Disarranges of the cardiovascular system are, as of now, recognized amid the starting examination of the infant, counting inadequate pieces of the proper bundle department of the His bundle, extrasystole, and signs of subendocardial ischemia. Subsequently, post-hypoxic cardiomyopathy is considered one of the hazard variables for the improvement of cardiovascular pathology (such as cadence unsettling influences, vascular dystonias, and others) in afterward stages of life [6,76,77,78].

We have established that modeling prenatal hypoxia in rats leads to a reduced heart rate and a critical dominance of parasympathetic innervation within the control of the heart’s electrical movement. The diminished heart rate after encountering hypoxia may well be caused by sinus square, which may also reflect the parasympathetic direction of the heart rather than thoughtful control of electrical movement beneath ordinary conditions. The improvement of unsettling influences within the bioelectrical movement of the heart after PH drove an expansion of the electrical systole of the ventricles, which may have been caused by impaired myocardial conductivity within the ventricles. Beneath these conditions, the control of ventricular electrical repolarization expanded 5.5 times, demonstrating noteworthy issues with the reclamation of the layer potential in ventricular cardiomyocytes [79].

Fetal hypoxia leads to a disturbance within the autonomic control of coronary vessels, deterioration of energy metabolism, and damage to the ultrastructure of mitochondria, both in cardiomyocytes and in the cells of the conduction system, which may be a possible cause of reduced myocardial contractility and impeded ordinary working of the sinoatrial node [79]. Within the contractile myocardium and conduction system during the post-hypoxic period, cells with signs of apoptosis and dystrophy are observed, with a certain correlation between the seriousness of morphological changes and the bioelectrical unsettling influences in beat and conductivity. The result of hypoxic myocardial damage can be focal dystrophy, which, if not adequately treated, may lead to focal cardiosclerosis [80,81].

To summarize the above, it can be said that PH is a powerful damaging factor that triggers mechanisms such as oxidative and nitrosative stress, mitochondrial dysfunction, disruption of energy supply to the heart, and apoptosis. Research into the molecular and biochemical mechanisms of post-hypoxic cardiomyopathy in newborns has shown that many of these processes are dependent on nitric oxide (NO). All of this makes the NO system an attractive area for study from the perspective of fundamental medicine and biology, as well as a promising target for pharmacological intervention.

### 2.2. Causes of Prenatal Hypoxia

PH is a consequence of a wide range of adverse processes occurring in the body of the child or the mother or in the placenta. The probability of PH increases in maternal diseases—anemia, cardiovascular pathology (heart defects, hypertension), kidney diseases, respiratory system diseases (chronic bronchitis, bronchial asthma, etc.), diabetes mellitus, pregnancy toxicosis, multiple pregnancy, and sexually transmitted infections. Alcoholism, nicotine, drug, and other types of addiction in the mother also influence the formation of PH. PH can develop due to disturbances in fetoplacental blood flow caused by threatened miscarriage, post-term pregnancy, pathology of the umbilical cord, fetoplacental insufficiency, abnormal labor, and other complications of pregnancy and childbirth. PH can be divided into preplacental hypoxia, uteroplacental hypoxia, and postplacental hypoxia [1,6,23,25,41,52,58,65,67,70,74].

### 2.3. The Role of NO in Heart Regulation

The role of NO in regulating various processes in the cardiovascular system is well known. Endothelial cells synthesize and discharge NO, which intercedes an assortment of impacts, counting vascular tone, hemostasis, blood pressure, and vascular remodeling [82,83,84,85]. The critical role of nitric oxide (NO) in cardiomyocyte function is well established, particularly in regulating ion channels, maintaining calcium (Ca^2+^) homeostasis, modulating contractility, supporting energy metabolism, influencing cell proliferation, and enhancing resistance to hypoxia [86,87]. NO exerts its metabolotropic, physiological, and other effects through various mechanisms. For instance, NO can post-translationally modify target proteins, primarily by adding a nitroso group to the thiol side chain of cysteine, a process known as S-nitrosylation, which leads to the acquisition of new properties by the protein [88,89]. However, the spatial range of NO’s direct actions is limited due to its short diffusion distance. Molecules of cellular carriers, such as S-nitrosoglutathione, intervene more in removed NO signal transmission, acting as a carrier and benefactor, exchanging NO to more far-off targets [90,91,92]. Interestingly, the administration of exogenous glutathione has been shown to modulate ventricular arrhythmias induced by mechanical stretch, suggesting a possible connection between NO signaling cycles and the mechanical responses of cardiomyocytes [93,94].

NO also activates the cGMP/protein kinase G (PKG)-dependent phosphorylation pathway. Enactment of this pathway leads to the phosphorylation of target proteins, restraint of the mitogen-activated protein kinase kinase/extracellular signal-regulated kinase (MEK1/2/ERK1/2) pathway, and actuation of the c-Jun N-terminal kinase (JNK)1, 2, and 3 pathways. The extreme result of this process is cardioprotection and the suppression of genes involved in hypertrophy, as well as the regulation of genes involved in apoptosis [95,96,97]. Data has been obtained indicating that in the myocardium, the cGMP nitrosylation pathway is mediated by eNOS (endothelial nitric oxide synthase) [89,98,99]. In healthy neonatal rat hearts, nitric oxide (NO) plays a regulatory role in the integrin complex. This complex consists of cytoskeletal proteins that are essential not only for mediating cell adhesion but also for sensing and transmitting mechanical stimuli. Integrins, which are heterodimeric transmembrane receptors, link the extracellular matrix to the actin cytoskeleton, thereby facilitating mechanical signal transduction to the cytoskeleton [100,101,102,103]. In cardiomyocytes, integrins contribute to maintaining cardiac function by modulating both mechanical and electrical coupling within the myocardium. In cardiomyocytes of healthy newborn rats, integrins, with the participation of NO, promote the release of Ca^2+^ from the sarcoplasmic reticulum [86,104]. Besides modulating calcium homeostasis via NO signaling, it has also been reported that NO can regulate integrin expression through the cGMP pathway [105]. The functional properties of certain ion channels in the myocardium can be regulated by NO through the nitrosylation of protein fragments of the channel (nitrosylation of cysteine fragments) [106]. This nitrosylation can either enhance or inhibit channel activity, depending on the specific ion channel involved [107]. Specifically, there are a few channels that are both mechanically delicate and balanced by NO through the nitrosylation of thiol bunches [108]. NO is discharged in cardiomyocytes in reaction to mechanical boosts and can control conductivity by modulating the movement of ion channels [109].

### 2.4. The Role of NO in the Heart During Fetal Development

The cardioprotective and endothelial-protective role of NO, produced by eNOS, is well known. NO provides protection against myocardial reperfusion injury, regulates the conduction system, participates in the synchronization of heart contraction and relaxation, activates compensatory energy shunts, modulates platelet aggregation, regulates vascular tone, and inhibits the proliferation of smooth muscle cells in blood vessels [110,111,112,113,114,115,116].

However, the global effects of NO on the developing cardiovascular system are not fully understood. It is known that NO influences the early migration of cardiac progenitor cells and vasculogenesis [117,118]. Nitric oxide stimulates soluble guanylate cyclase, promoting the formation of cyclic GMP (cGMP), a key secondary messenger that regulates various protein targets, including bone morphogenetic protein-4 (BMP4). BMP4 contributes to the positioning of the heart during embryonic development by guiding the migration of cardiac progenitor cells toward the embryo’s left side [117,119]. Additionally, NO may modulate BMP4 signaling through the production of reactive nitrogen species, such as peroxynitrite [120]. NO causes organ transposition by altering the migration of cardiac progenitor cells from blood islands. NO regulates the expression of heart-specific genes and also affects apoptotic signaling [121]. NO promotes cardiac differentiation by both switching towards a cardiac phenotype and inducing apoptosis in cells not committed to cardiac differentiation [122].

It is also known that iNOS and, especially, eNOS are significantly expressed during the early stages of cardiomyogenesis. Reduced expression of eNOS inhibits the maturation of terminally differentiated cardiomyocytes [123]. NO positively regulates the expression of genes involved in heart morphogenesis, as well as controlling heart contraction, cardiac cell development, calcium signaling, and the structure and development of the heart in the embryo [124,125,126].

### 2.5. Changes in the Nitric Oxide System in the Heart of Offspring After PH

It has been established that alterations in nitric oxide levels during pregnancy are associated with the development of classical symptoms of eclampsia, disturbances in placental formation, modifications in placental blood flow, embryopathy, fetopathy, intrauterine growth restriction, and fetal demise [127]. Nitric oxide serves dual roles as both a factor in disease pathogenesis and as a protective agent at the cellular and organ levels, including cardioprotective and endothelioprotective functions [128,129]. Additionally, NO is crucial for endothelial cell development and acts as a key regulator of the vascular endothelial growth factor (VEGF) family, which includes placental growth factor (PGF), angiopoietins (ANG-1 and ANG-2), and their soluble receptors (sFlt-1 and Tie-2) [130]. The VEGF family plays an essential role in proper placental vascularization, angiogenesis, and remodeling throughout pregnancy [131,132].

NO production increases pro-angiogenic VEGF-A and PGF in human trophoblast cultures, whereas inhibition of NO synthesis leads to elevated SFLT-1 levels and hypertensive reactions in pregnant mice [133,134,135]. Nitric oxide additionally downregulates the expression of endothelial adhesion molecules and pro-inflammatory cytokines and can quickly induce HIF-1α expression [136,137,138]. Under conditions of prenatal hypoxia, on the one hand, NO production increases, while on the other hand, the synthesis of essential factors for the preservation and transport of this molecule decreases, leading to NO lack within the heart and blood vessels. Amid delayed pre-birth hypoxia, ROS can influence NO bioavailability [139]. Increased ROS levels and an imbalance in the ROS/NO ratio in newborns after prenatal hypoxia contribute to enhanced peripheral vasoconstriction, causing hypoxic damage to vital organs, including the heart and brain [140].

PH also enhances the expression of iNOS mRNA and increases iNOS protein levels in the ventricles of the fetal heart [141]. These findings were confirmed by clinical studies, which demonstrated that hypoxia decreases eNOS action and quality expression within the cardiac tissue of patients with cyanotic inherent heart surrenders. In differentiation, iNOS action and expression are expanded in cyanotic children [142]. Prenatal hypoxia also elevates NADPH oxidase 1 homolog expression, promoting superoxide generation, which reacts with NO to produce peroxynitrite, thereby diminishing NO bioavailability [143]. Long-term alterations in the cardiac nitric oxide system following experimental prenatal hypoxia include downregulation of eNOS mRNA and protein, upregulation of iNOS mRNA and protein, decreased NO availability, and enhanced nitrosative stress [144]. This is also confirmed by other studies, which show a decrease in eNOS mRNA and protein in the cardiomyocytes of adult animals exposed to PH [145]. Apparently, there is a reduction in NO production by endothelial nitric oxide synthase, while the NO produced during iNOS expression activation is converted into peroxynitrite. The obtained results indicate significant disruptions in the NO system of the myocardium in rat pups after PH—changes in the expression pattern of NOS, reduced NO bioavailability, and activation of nitrosative stress.

NO deficiency leads to a number of serious disturbances in the offspring’s organism after PH [11]. Such changes in the nitric oxide system of the myocardium in the offspring after PH align with current views on the mechanisms of myocardial damage in ischemia and hypoxia, as developed through experimental studies and clinical observations [142,146]. It is well known that PH impairs the heart’s tolerance to ischemia/reperfusion, damages endothelial-dependent mechanisms of vasodilation/vasoconstriction, and further contributes to the development of cardiovascular pathologies such as hypertension, atherosclerotic vascular diseases, and congestive heart failure, which occur in the presence of NO deficiency [147].

Data indicate that intrauterine hypoxia leads to decreased expression and activity of endothelial nitric oxide synthase (eNOS) in both cardiomyocytes and endothelial cells, contributing to a higher likelihood of endothelial dysfunction. The impaired function of eNOS may result from altered interactions with its regulatory partners, including caveolin-1, calmodulin, and Hsp90. Modifications within the phosphorylation and dephosphorylation of key serine and threonine buildups in eNOS may also be included within the brokenness of eNOS movement [1,148]. Disturbances in endothelial-dependent vasodilation have been recognized within the coronary supply routes of both male and female siblings uncovered to PH at the ages of 4 and 9.5 months, against the foundation of diminished eNOS and disabled work of SKCa and IKC channels [26]. Low levels of eNOS lead to the disruption of NO-dependent regulation of glutathione synthesis and a reduced resistance to oxidative stress [149]. The decrease in eNOS may occur due to a deficiency of HIF-1α, as this factor activates eNOS expression by phosphorylating the serine residue [150].

Taking after a diminishment in eNOS after PH, there is a raised expression of iNOS, which serves to compensate for the diminished NO generation [151]. PH increases the expression of inducible nitric oxide synthase (iNOS) mRNA and iNOS protein levels within the ventricles of the fetal guinea pig heart [141]. Elevated iNOS activity can be associated with cofactor deficiencies and the generation of superoxide and other reactive nitrogen species [152]. Be that as it may, beneath conditions of diminished thiol cancer prevention agents, this leads to the arrangement of cytotoxic NO subsidiaries in “parasitic responses”.

Similarly, studies have shown that in preeclampsia, decreased endothelial NO synthesis and redox-mediated conversion of NO to peroxynitrite cause reduced systemic NO concentrations [128]. Such responses can happen in conditions of L-arginine insufficiency, antioxidant inadequacy, mitochondrial brokenness, and expanded iNOS expression. The uncontrolled arrangement of cytotoxic NO derivatives leads to the nitration of the foremost dynamic locales in protein structures, particle channels, receptors, transmembrane pores, and signaling particles, i.e., the development of nitrosative stress.

A similarly imperative result of myocardial ischemia is the loss of NO-mediated effects, such as the restraint of cell multiplication, platelet aggregation, and, most critically, the concealment of monocyte enactment by so-called attachment particles [153]. Nitrosative stress also causes depletion of heat shock protein 70 (HSP70) within cells, exacerbated by impaired glutathione-dependent thiol-disulfide homeostasis. Cytotoxic shapes of NO not as it was adjusted (both reversibly and irreversibly) macromolecules, counting HSP70 itself, but moreover decreased the expression action of qualities encoding its amalgamation [154,155]. The part of NO derivatives in stifling gene action and lessening the levels of different translation variables has been illustrated [156]. It is evident that elevated levels of nitrogen oxides like peroxynitrite and nitrosonium initially nitrate thiol–redox-sensitive regions of these genes and, with increasing concentrations, subsequently oxidize them [157].

## 3. Mechanisms of Cardiovascular Dysfunction After Prenatal Hypoxia

### 3.1. Disruption of Energy Metabolism in the Myocardium and Mitochondrial Dysfunction in the Offspring After PH

The structural integrity and functional activity of myocardial mitochondria ensure its main role as the cellular “power station”—producing energy to sustain cardiac function [158,159,160]. Disruption of mitochondrial functional activity leads to various functional and pathological shifts in heart function (Figure 2). The concept of mitochondrial dysfunction has become a general pathological term. Mitochondrial dysfunction, whether primary or secondary in origin, plays a significant role in the pathogenesis of numerous cardiovascular diseases. It contributes to the development of hypertrophic cardiomyopathy, exacerbates reperfusion injury following myocardial ischemia, and is involved in diastolic dysfunction that can progress to heart failure. Additionally, mitochondrial impairment is linked to the early onset of heart failure and is associated with hypertension during pregnancy in women [161,162,163,164]. A decrease in the functional activity of myocardial mitochondria leads to ATP deficiency and disruption of the heart’s energy metabolism [115,165,166]. Impaired ATP transport results in a deficiency of creatine phosphate, suppression of the creatine phosphate shuttle mechanism, and disruption of the energy capacity required for rapid support of heart contraction. All of this forms the basis for the development of heart failure [167,168].

It is known that in the heart of adult offspring after PH, there is a persistent disruption in the activity of enzymes and the expression of proteins in mitochondrial complexes of oxidative phosphorylation under hypoxia [169]. The succinate dehydrogenase complex, a key component of the Krebs cycle within mitochondria, is particularly vulnerable to prenatal hypoxia (PH). Alterations in cardiac developmental programming induced by PH can disrupt this succinate-dependent pathway in female fetuses, potentially resulting in lasting impairments in mitochondrial function within the hearts of young adult female offspring [170].

PH is also known to impair electron transfer between cytochrome c and oxygen within cardiac mitochondria, contributing to mitochondrial dysfunction. This impairment manifests as reduced stroke volume and cardiac output in male offspring [171]. Additionally, PH downregulates the expression of mitochondrial transcripts, including peroxisome proliferator-activated receptor gamma coactivator 1-alpha (PGC-1α), cytochrome c oxidase subunit II (COXII), and uncoupling proteins. This reduction compromises mitochondrial respiratory efficiency in the hearts of affected offspring [172]. It is known that NO can serve as a natural short-term regulator of mitochondrial physiology, enhancing the efficiency of oxidative phosphorylation in redox processes by reducing errors and failures in proton pumps [173]. Therefore, its deficiency due to PH may lead to disruptions in oxidative processes in mitochondria.

Cytotoxic products of nitric oxide metabolism play a direct role in the formation of mitochondrial dysfunction under PH conditions—reducing the energy-producing function of mitochondria, inhibiting complexes I/IV, and progressing the decline in energy metabolism [174]. These findings align with research indicating that NO generated by inducible nitric oxide synthase (iNOS) is converted to peroxynitrite, which inhibits mitochondrial oxidative phosphorylation. Peroxynitrite irreversibly nitrosylates electron transport chain enzymes and depletes iron, impairing respiration, lowering membrane potential, and triggering mitochondrial dysfunction and apoptosis [175,176].

Evidence also suggests that NO can directly induce the opening of the mitochondrial permeability transition pore (megachannel), promoting cytochrome c release and activating the caspase cascade. Moreover, NO and its derivatives (such as peroxynitrite and the nitrosonium ion) oxidize thiol groups on mitochondrial membrane proteins, facilitating the cytosolic release of pro-apoptotic factors [11,177,178,179]. Suppression of protein synthesis in mitochondria after hypoxia is associated with an increased susceptibility of cells to apoptotic stimuli mediated by NO [180,181,182].

In cases of insufficient antioxidant system activity, a vicious cycle may form between the production of ROS by the mitochondrial electron transport chain and the mutational process in mitochondrial DNA. The consequence of this is the progression of mitochondrial dysfunction after PH [183,184,185]. Disruption of any mitochondrial function—energy-producing, death-inducing—or activation of ROS production by mitochondria can serve as a cause for the development of functional and morphological heart abnormalities in the offspring after PH [170,183,186].

The mitochondrial apparatus of ventricular cardiomyocytes in rats after PH is characterized by pronounced degradation processes in the organelles of the subsarcolemmal zone, swelling of “high-energy” mitochondria in the intermyofibrillar zones, and “low-energy” mitochondria in the perinuclear zone of cardiomyocytes, with a reduction in the number of associations between mitochondria. Myofibrils appeared fragmented, and mitochondria varied in size. Lipid inclusions were found in the sarcoplasm. In the contractile cardiomyocytes after PH, areas of myofibrillar apparatus over-contraction, known as “rigor”, were observed. Ischemic but minimally altered heart cells are often referred to in the literature as “oscillating”, as they are in a state of electrical instability [79,187,188].

The main ultrastructural manifestation of this is the phenomenon of myofibril contracture, recorded during electron microscopy studies of experimental animals after PH, which is an undeniable confirmation of both hypoxic and ischemic cell damage. The latter is a pathognomonic sign of disrupted sarcolemma permeability and the movement of Ca^2+^ ions from the intercellular space into the cardiomyocyte, indicating the development of ion imbalance. Evidence of this is also the appearance of electron-dense inclusions in the mitochondria and relative expansion of the sarcoplasmic reticulum. These cells can serve as a source of rhythm disturbances [171,188,189,190].

Moreover, mitochondrial dysfunction can lead to the activation of the immune system. The production of ROS from dysfunctional mitochondria can result in damage to lipids and proteins, which may activate inflammatory pathways [191,192,193,194]. Dysfunctional mitochondria can release damage-associated molecular patterns, including cardiolipin, N-formyl peptides, ROS, and mtDNA, which can activate the inflammasome [195,196,197]. Activation of the immune system and increased expression of pro-inflammatory cytokines lead to further upregulation of iNOS expression, increasing the production of NO and ROS, which can result in further detrimental effects on already dysfunctional mitochondria [198,199,200].

### 3.2. Nitrosative Stress in the Heart of Offspring After PH

Several studies have shown that experimental PH leads to dysfunction of the nitric oxide system in the myocardium of both the fetus and the offspring. The authors have demonstrated that PH suppresses the expression and activity of endothelial nitric oxide synthase (eNOS) and significantly increases the inducible form (iNOS), along with increased NO production. In this situation, NO loses its physiologically “beneficial” properties and is converted into peroxynitrite and other cytotoxic forms [201,202,203]. Most of the cytotoxic effects of NO actually belong to ONOO^−^, which is formed in a reaction with superoxide (O_2_^−^). Indeed, peroxynitrite is significantly more active; it intensely nitrosylates proteins and can be a source of the highly toxic hydroxyl radical ^•^OH. Under normal physiological conditions, a balance between superoxide and nitric oxide exists in vivo. NO and superoxide react together at a diffusion-controlled rate, forming peroxynitrite (ONOO^−^), which causes cell damage by oxidizing many biological molecules. In addition, ONOO^−^ participates in the inactivation of Mn-SOD and Fe-SOD [11,111,115,141]. Intemperate generation of NO can actuate nitrosative stress, leading to the arrangement of peroxynitrite, which may increment the expression of matrix metalloproteinases (MMPs). PH causes oxidative–nitrosative stress and alters the expression of extracellular matrix proteins through the regulation of the iNOS pathway in the fetal heart ventricles. This characterizes NO, produced by iNOS, as a key stimulus for initiating the adverse effects of peroxynitrite in the fetal heart [141,201].

PH promotes the accumulation of peroxynitrite in the myocardium of offspring by elevating inducible nitric oxide synthase (iNOS)-derived NO production. Peroxynitrite subsequently activates the nuclear translocation of transcription factors, including nuclear factor kappa B (NF-κB) and activator protein-1 (AP-1), which enhance the transcription of matrix metalloproteinase (MMP) genes [204,205]. The MMP gene family comprises various enzymes responsible for remodeling the extracellular matrix (ECM), with MMP2 and MMP9 being particularly important in the structure and function of cardiac tissue. The involvement of MMPs in ECM modulation within the myocardium is well known in cardiac pathologies (e.g., heart disappointment, postnatal hypoxic cardiomyopathy, ischemia-reperfusion damage, and contractile dysfunction of the myocardium) [204,206].

The elevated expression and activation of MMP2, MMP9, and MMP13 in response to reactive nitrogen species promote excessive collagen synthesis, disturbing the equilibrium between ECM synthesis and degradation and ultimately contributing to collagen deposition and myocardial fibrosis [207,208,209]. Peroxynitrite, a highly reactive nitrogen species, exerts multiple downstream effects, including the regulation of gene expression, protein oxidation, and nitration, induction of DNA damage, and promotion of lipid peroxidation [210,211,212]. Additionally, peroxynitrite augments MMP9 enzymatic activity by inducing autolytic cleavage of cysteine thiol groups and enhances MMP9 gene expression by activating nuclear factor kappa B and activator protein-1 transcriptional pathways [213,214,215].

Currently, there is a generalized concept of “nitrosative stress”. As a result of the action of reactive forms of NO, either nitrosylative stress (formation of nitrosamines, S-nitrosothiols, deamination of DNA bases) or oxidative stress develops [11,216]. NO forms reactive intermediates such as nitrosonium (NO^+^), nitroxyl (NO^−^), and peroxynitrite (ONOO^−^). Most of the cytotoxic effects of NO actually belong to ONOO^−^, which is formed in a reaction with superoxide (O_2_^−^). Peroxynitrite is significantly more active; it intensely nitrosylates proteins and can be a source of the highly toxic hydroxyl radical (^•^OH) [217,218]. NO^+^ is a powerful nitrosylating agent, and its targets can be nucleophilic groups of active thiols, amines, carboxyls, hydroxyls, and aromatic rings. NO^+^ is formed from excess NO with the participation of divalent iron and oxygen [216,219]. NO^−^ has reducing properties and exerts positive inotropic and lusitropic effects on the myocardium. In ischemia or hypoxia, NO^−^ in the conditions of developing lactate acidosis exhibits pro-oxidant properties towards thiols and amines. It has been shown that NO^−^ decreases glutathione levels and disrupts electrical activity, inhibiting sodium channel activity in the heart [11,220].

Apparently, the dual effect of NO^−^ is related to its concentration, as an increase in its levels leads to the formation of the toxic nitrite anion. N_2_O_3_, being a source of NO^+^, exhibits strong nitrosylating properties, interacting with aliphatic and aromatic amines to form N-nitrosamines. Nitrosamines, and specifically their conversion products under the action of P450 enzymes (such as diazonium ions and formaldehyde), are alkylating agents for nucleic acids, deaminating purines, inhibiting O⁶-methylguanine-DNA methyltransferase, and increasing the formation of 8-hydroxyguanine. N_2_O_3_ reacts with cysteine to generate S-nitrosocysteine and glutathione to form S-nitrosoglutathione, which serves as the primary carrier for NO transport [221,222,223,224,225].

Research indicates that NO can be transported via N_2_O_3_-mediated nitrosylation of thiol groups, with subsequent NO release facilitated by disulfide isomerase activity [226,227,228]. Another pathway for NO release involves the enzymatic action of glutamyltranspeptidase on S-nitrosoglutathione, generating S-nitrosocysteinylglycine, which then serves as a source of NO. Cystine participates in the transport of S-nitrosoglutathione, being reduced to cysteine, which then reacts with S-nitrosoglutathione to form S-cysteine. S-cysteine plays a role in rapid signal transmission, forming adaptive responses to hypoxia [229,230,231,232]. Glutathione reductase and glutathione transferase regulate these processes. Under ischemic conditions, inhibition of these enzymes promotes oxidative modification of low-molecular-weight thiols, causing homocysteine accumulation and impairing NO transport, which fosters the formation of cytotoxic NO derivatives and amplifies thiol oxidation [233,234].

An adequately functioning thiol-based antioxidant system is critical for maintaining proper NO transport and ensuring cellular resistance to nitrosative stress. Within mitochondria, NO transiently binds to cytochrome c oxidase, thereby inhibiting oxidative phosphorylation. This disruption of electron transport promotes superoxide production and subsequently facilitates peroxynitrite (ONOO^−^) formation [235,236,237]. Peroxynitrite generation is particularly prominent in cells exhibiting elevated inducible nitric oxide synthase (iNOS) activity and heightened reactive oxygen species (ROS) production via enzymes such as xanthine oxidase, NADH oxidoreductase, cyclooxygenase, lipoxygenase, and components of the electron transport chain. Peroxynitrite exerts its nitrosative effects on a wide array of molecular targets, including thiols, carbon dioxide, metalloproteins, nucleic acids, metabolite-related signaling molecules, and membrane lipids [238,239].

Peroxynitrite, although relatively stable, undergoes rapid protonation under acidic pH, predominantly forming nitrate anions along with hydroxyl radicals and nitrogen dioxide, which drive its oxidative reactivity. It disrupts the metabolic interplay between methionine and cysteine by inhibiting critical enzymes regulating cysteine homeostasis, thereby promoting homocysteine accumulation. Additionally, peroxynitrite reacts with carbon dioxide to produce a potent nitrosylating species—nitrosoperoxocarbonate. A key neurotoxic mechanism of peroxynitrite involves its reaction with tyrosine residues, resulting in nitrotyrosine formation. Peroxynitrite impairs the activity of Cu-Zn-superoxide dismutase (SOD) and Mn-SOD through nitration of tyrosine-34 and by binding to copper ions, thereby altering their oxidation state [240,241,242,243,244].

This reactive species irreversibly inhibits mitochondrial respiration during ischemia by interacting with iron centers of enzymatic active sites and by nitrosylating sulfur-, nitrogen-, and oxygen-containing groups (thiol, phenolic, hydroxyl, amine) within enzyme proteins. In cases of more pronounced nitrosative stress, it irreversibly oxidizes them [245,246]. The spectrum of peroxynitrite activity also includes the nitrosylation of guanine and DNA strand breaks, which can lead to mutations or trigger apoptosis. In relation to genomic damage, another effect of NO is known: products of its reaction with O_2_ inhibit enzymes responsible for DNA repair. Depending on the source (different NO donors), the effects of NO on alkyltransferase, formamidopyrimidine-DNA glycosylase, and ligase have been shown. It is also known that NO can activate PARP- and ADP-ribosylation, possibly due to DNA breaks, but this more likely leads to necrosis due to depletion of NAD and ATP pools [247,248,249,250].

Evidence suggests that nitric oxide (NO) directly triggers the opening of the mitochondrial permeability transition pore (MPTP), resulting in cytochrome c release and subsequent activation of the caspase cascade [11,251]. These findings were obtained by treating mitochondria with cytotoxic NO derivatives, including peroxynitrite and the nitrosonium ion, which modify thiol groups of proteins within the mitochondrial pore complex [252]. NO and its reactive derivatives promote peroxidative damage to mitochondrial phospholipids. Exposure to cytotoxic NO derivatives and hydroxyl radicals induces the opening of mitochondrial pores, facilitating the release of pro-apoptotic proteins into the cytosol. This pore opening is mediated by oxidation or nitrosylation of thiol groups within cysteine residues of the inner mitochondrial membrane ATP/ADP antiporter, transforming it into a nonspecific, permeable channel. As a result, mitochondria shift from efficient energy producers (“powerhouses”) to sites of uncontrolled substrate oxidation lacking ATP production (“furnaces”).

Our research has demonstrated profound alterations in the myocardial NO system of rats following PH, marked by an imbalance in eNOS/iNOS expression, reduced NO bioavailability, and elevated nitrotyrosine levels [115,144,252], amid the inhibition of glutathione-dependent enzymes GPX1 and GPX4 activity [253]. An equally critical consequence of hypoxia-induced nitrosative stress in the myocardium is the loss of NO-dependent functions, including inhibition of cell proliferation, platelet aggregation, and, importantly, suppression of monocyte activation via adhesion molecules [153]. Nitrosative stress further results in HSP70 deficiency, occurring alongside glutathione depletion within the thiol-disulfide regulatory system. Cytotoxic NO derivatives not only modify (both reversibly and irreversibly) various macromolecules, including HSP70 but also downregulate the transcription of genes responsible for HSP70 synthesis [154,155]. We have established that PH leads to a decrease in HSP70 expression against the background of suppressed eNOS expression and a significant increase in nitrotyrosine concentration [254]. The part of NO subordinates within the concealment of gene movement and the reduction of various transcription factor levels has been demonstrated [156]. Apparently, an excess of such forms of nitric oxide as peroxynitrite and the nitrosonium ion initially nitrosylate thiol-redox-dependent regions of these genes and then, with increasing concentration, oxidize them [157].

### 3.3. NO-Dependent Mechanisms of Endothelial Dysfunction After PH

Cardiovascular disorders caused by hypoxia are a major cause of illness in newborns, with studies showing that 40–70% of children who have undergone intrauterine hypoxia experience these issues. These disorders play a significant role in the development of numerous, often severe, diseases in both children and adults [1,4,5]. The mechanisms behind post-hypoxic heart disorders are not yet fully understood, making them a significant issue in pediatric cardiology. The clinical symptoms of this pathology during the acute phase are polymorphic, often resembling other diseases, and there is frequently a need for differential diagnosis with congenital heart defects, congenital carditis, and cardiomyopathies [36,255].

According to current understanding, endothelial dysfunction, along with disruptions in the NO system, serves as a key underlying factor in the pathogenesis of various cardiovascular diseases [12,148]. Prenatal hypoxia leads to asymmetric fetal growth restriction, hypertrophic remodeling of the heart and aorta, altered cardiac function, and increased sympathetic innervation of peripheral resistance arteries in neonates. In later life, prenatal hypoxia contributes to the emergence of hypertension, ischemic heart disease, heart failure, metabolic syndrome, and heightened susceptibility to ischemic injury [23,256]. The presence of endothelial dysfunction mechanisms has been identified in cardiovascular pathology following PH. Clinical signs of impeded utilitarian state and maladaptation of the cardiovascular system after prenatal hypoxia directly correlated with signs of endothelial dysfunction (changes in the production of endothelin-1, NO, VEGF, circulating desquamated endothelial cells) in both newborns and in later stages of life [68,257,258,259].

Endothelial dysfunction and cardiovascular pathology, including those following prenatal hypoxia, are partially attributed to disturbances in the nitric oxide system. Research indicates that prenatal hypoxia affects both the synthesis and bioavailability of NO. During hypoxia in the prenatal period, increased levels of superoxide radicals and other reactive oxygen species can lead to oxidative modification of NO, converting it into peroxynitrite, which has a damaging effect on fetal organs [23,258,260]. Hypoxia reduces the expression of eNOS and can also affect its enzymatic activity by modulating its post-translational modifications. Under hypoxic conditions, when there is a deficiency of L-arginine, eNOS may produce superoxide radicals instead of NO. These impairments in eNOS function are regarded as a central mechanism underlying endothelial dysfunction in cardiovascular pathologies [11,68].

We have identified a decrease in eNOS expression alongside a significant increase in nitrotyrosine levels in both 1-month-old and 2-month-old rats after prenatal hypoxia. This is likely due to increased ROS generation during prenatal hypoxia, which diminishes NO bioavailability and suppresses eNOS expression [254]. Elevated NADPH levels during prenatal hypoxia promote ROS formation, which interacts with NO to produce stable peroxynitrite anions, further lowering NO bioavailability [246].

PH significantly reduces NO-dependent vasodilation mediated by acetylcholine in the thoracic aortic rings of both fetal and adult offspring. In the offspring after PH, there is reduced expression of eNOS, primarily due to increased expression of NADPH oxidase type 2 and high levels of ROS production against the background of elevated miR-155-5 levels in endothelial cells of blood vessels [261,262]. NADPH oxidase type 2 is a key component in the generation of ROS after PH, which makes NO more susceptible to oxidation, reducing its stability and initiating the first step toward the formation of endothelial dysfunction [82,263].

The excess of ROS during hypoxia removes NO produced by eNOS in endothelial cells, thereby limiting the bioavailability of NO. ROS impairs NO-mediated dilation of coronary microvessels by increasing the activity of arginase. The formed peroxynitrite oxidizes tetrahydrobiopterin (BH4), a cofactor required for eNOS, leading to eNOS uncoupling. Additionally, oxidative stress disrupts the balance between L-arginine and asymmetric dimethylarginine (ADMA) [68]. The expression of arginase in endothelial cells during hypoxia is induced by the activated ROS through the RhoA/Rho kinase pathway [264,265]. Chronic prenatal hypoxia results in a reduced expression of HIF-1 mRNA in cells of various organs in rats [266], which, in our view, may reflect the depletion of compensatory-adaptive mechanisms following intrauterine hypoxia. Hypoxia-inducible factors (HIFs) function as transcription factors that control the expression of genes responsible for synthesizing proteins involved in physiological responses to hypoxia or ischemia [267]. Under hypoxic conditions, hypoxia-inducible factors (HIFs) exert cytoprotective functions by stimulating repair mechanisms and increasing the expression of antioxidant enzymes and factors, including heme oxygenase-1, VEGF, and angiopoietins [268]. HIF-1 plays a key role in adapting cellular energy metabolism to hypoxia by modulating compensatory ATP production pathways, promoting glutathione biosynthesis, and enhancing cellular defenses against oxidative stress. HSP70 is known to stabilize HIF-1 functionality, thereby prolonging its activity under stress conditions. Our observations indicate that intrauterine hypoxia-induced downregulation of HIF-1 mRNA is accompanied by a concurrent reduction in HSP70 levels. Multiple studies have reported that HIF-1 levels and its isoform expression profiles vary depending on the severity, duration, and tissue-specific context of hypoxic exposure [265,268,269]. In the presence of nitrosative stress and the accumulation of cytotoxic NO metabolites coupled with ATP depletion, HIF levels are reduced as a result of enhanced ubiquitin-independent degradation of oxidatively modified HIF-1α and suppression of its synthesis under energy-deficient conditions. The regulatory role of nitric oxide in modulating HIF-1α mRNA expression is well documented [138].

Our experimental data provide strong evidence that modeled prenatal hypoxia significantly impairs cardiovascular function in the offspring (rats aged 1 and 2 months). In the myocardium of rats exposed to prenatal hypoxia, we recorded elevated levels of the endothelial dysfunction marker sEPCR, along with decreased levels of Tie-2 and VEGF-B, which serve protective roles, as well as a notable antioxidant deficiency (manifested as reduced Cu/ZnSOD and GPX) [253]. The disturbances observed in the nitric oxide system following prenatal hypoxia, along with elevated specific protein markers, indicate impaired ischemia/reperfusion tolerance, disrupted endothelial regulation of vascular tone, and further progression of endothelial dysfunction resulting from intrauterine hypoxia.

The development of endothelial dysfunction following prenatal hypoxia (PH) occurs in the context of HIF-1α deficiency—a key factor that promotes eNOS expression through serine phosphorylation—as well as nitrosative stress. This stress leads to a reduction in HSP70 levels, depletion of the glutathione-dependent thiol-disulfide system, diminished NO bioavailability, and inhibition of gene transcription due to cytotoxic NO derivatives [144,155,254]. Our findings [144,253,254] indicate that PH causes pathological alterations in the cardiovascular system of neonates and contributes to the onset of endothelial dysfunction. It is established that EPCR expression is upregulated on endothelial cells during post-ischemic neovascularization. Notably, exogenous NO administration significantly enhances the formation of angiogenic endothelial sprouts from both aortic rings and primary endothelial cells isolated from PAR1-mutant mice. This suggests that maintaining NO bioavailability during angiogenesis is a crucial function of endothelial signaling via the EPCR-PAR1 axis [270,271].

We also observed a downregulation of major antioxidant enzymes, accompanying the previously reported elevation of nitrotyrosine levels in the myocardium of 1- and 2-month-old rats post-PH [144]. These findings indicate a marked intensification of oxidative stress following PH. Oxidative stress in the fetal cardiovascular system serves as a fundamental mechanism by which PH programs future cardiovascular diseases and endothelial dysfunction [256]. Our results align with other studies demonstrating that PH leads to aortic wall thickening, elevated nitrotyrosine staining, increased cardiac HSP70 expression, impaired NO-mediated vascular relaxation, and heightened myocardial contractility with a predominance of sympathetic tone [36].

GPX-4 is a key enzyme responsible for protecting cells during oxidative stress. It catalyzes the reduction of lipid hydroperoxides, including those incorporated into cellular membranes and lipoproteins. GPX-4 is capable of reducing hydroperoxides of various substrates, such as fatty acids, cholesterol, and thymine derivatives. This enzymatic activity is crucial for maintaining membrane integrity by preventing lipid peroxidation and thereby protecting cells from oxidative damage. Additionally, GPX-4 is pivotal in preventing ferroptosis, a distinct iron-dependent cell death process driven by lipid peroxidation and ROS accumulation [272,273].

GPX-4 is also required for maintaining mitochondrial viability by reducing cardiolipin hydroperoxides, key mediators of mitochondrial membrane destabilization. It exerts a direct antioxidant effect on membrane lipids, thereby serving as a crucial inhibitor of ferroptosis induced by lipid peroxide accumulation. The cytoplasmic isoform of GPX-4 plays a dominant role in ferroptosis suppression in somatic cells, whereas the mitochondrial isoform (mGPX-4) contributes to the prevention of mitochondrial dysfunction [274]. Recent findings have shown for the first time that prenatal hypoxia (PH) can trigger ferroptosis in human trophoblast cells, potentially contributing to miscarriage and underscoring the critical protective role of GPX-4 in this context [275].

GPx-1, another intracellular antioxidant enzyme, converts H_2_O_2_ into water, limiting oxidative damage and regulating H_2_O_2_-mediated signaling pathways tied to growth factors, mitochondrial function, and redox homeostasis. The decrease in GPx-1 expression in the myocardium of rats post-PH, as observed in our previous study [144], may be linked to an overproduction of cytotoxic NO forms amid high iNOS expression [276]. GPx-1 is key in preserving endothelial function and maintaining NO bioavailability [276], and its deficiency results in significant vasoconstriction and contributes to the development of endothelial dysfunction [277].

Superoxide dismutases (SODs) are generally divided into four categories: manganese SOD (MnSOD), copper–zinc SOD (Cu/ZnSOD), iron SOD (FeSOD), and nickel SOD (NiSOD). Among them, Cu/ZnSOD and MnSOD are found in the cytoplasm and are considered the primary enzymes responsible for eliminating free radicals inside the cell. These forms of SOD have drawn significant interest due to their crucial physiological roles and potential therapeutic applications [278]. In our studies, we observed reduced levels of Cu/ZnSOD in the cytosol of rats exposed to prenatal hypoxia (PH), a finding consistent with other research showing that PH suppresses Cu/ZnSOD expression at both transcriptional and post-translational stages. PH also lowers the enzymatic activity of Cu/ZnSOD, which may contribute to the development of cardiovascular diseases [279] and endothelial dysfunction [280].

Substantial evidence supports the association between diminished activity of antioxidant enzymes and negative pregnancy outcomes. Oxidative stress adversely affects maternal physiology, the progression of pregnancy, and fetal development. It interferes with placental function, hinders the transfer of oxygen and nutrients to the fetus, and leads to impairments in the cardiovascular system—especially in the form of cardiomyopathy and endothelial dysfunction [281].

### 3.4. NO and Cardiomyocyte Apoptosis After PH

Since hypoxia is a powerful stress factor, many researchers show that perinatal hypoxia induces cell death by activating both apoptotic and necrotic pathways, depending on the cell type [282,283,284,285]. The study of molecular mechanisms of apoptosis in various forms of heart pathology is one of the pressing issues in medical science. For a long time, apoptosis was considered atypical for highly differentiated tissues. However, in recent years, cardiomyocyte apoptosis after PH has been identified. At the same time, the specific features of the induction and course of cardiomyocyte apoptosis during and after intrauterine hypoxia are not yet fully understood [286,287].

It is known that apoptosis is programmed cell death, which, unlike necrosis, is an active and highly regulated process. It involves a cascade of specific signaling and effector molecules that interact with each other with a high degree of selectivity and sequence [286,288,289]. As a result, cell and nuclear shrinkage occurs, along with DNA fragmentation, chromatin condensation, and the subsequent formation of “apoptotic bodies”, which are membrane-bound clusters of condensed cellular contents that the cell breaks down into during apoptosis. These “apoptotic bodies” are either phagocytosed or degrade with subsequent breakdown (secondary necrosis). However, in both cases, an inflammatory response does not develop [287,290,291,292].

Programmed cell death plays a role in postnatal morphogenesis of the heart’s conduction system: the sinoatrial and atrioventricular nodes, as well as the His bundle [293,294]. Apoptosis of pacemaker cells may play a role in the development of paroxysmal arrhythmias, conduction disturbances, and the genesis of sudden coronary death. For cells that have reached terminal differentiation, such as cardiomyocytes, apoptosis is not typically characteristic. Both internal (controlled by mitochondrial activity) and external (initiated by death receptors) apoptotic pathways jointly regulate the mechanisms of heart development. During heart development, many cell populations are recruited into the heart, where they differentiate into cardiomyocytes, fibroblasts, smooth muscle cells, endocardial and endothelial cells lining the inner surfaces, and epicardial cells lining the outer contours. Thus, cell populations originating from the neural crest, which migrate to specific sites in the heart, are prone to apoptosis [33,35,38,291,292]. However, in cardiomyopathies, myocardial hypertrophy, and chronic heart failure of various etiologies, there is often a progressive decline in the contractile ability of the left ventricle. This process frequently occurs in the absence of any signs of myocardial ischemia. Therefore, apoptosis of cardiomyocytes has been used as a working hypothesis to explain the mechanism of heart failure development, which is supported by several experiments [293,295,296,297,298,299].

In the early stages of ischemia, apoptosis is the predominant form of cardiomyocyte death in newborns. A typical response of apoptosis in hypoxic cardiomyopathies and congenital heart defects is the intensification of its mitochondrial pathway. Heart failure resulting from perinatal hypoxia is accompanied by the activation of all pathways of programmed cell death, with the extent of apoptosis induction depending on the stage of circulatory failure [300,301,302]. The dynamics of apoptosis markers—lymphocytes with activated CD95+ expression in the blood of patients with chronic heart failure—can be used as a criterion for evaluating the effectiveness of therapy [303,304,305].

The initiation of apoptosis in cardiomyocytes under hypoxia can be triggered by a variety of stimuli. However, all these activation pathways converge on the activation of the aspartate-specific cysteine protease system, known as caspases, which are constitutionally expressed in cells as inactive zymogens. Once, under the influence of apoptosis inducers, caspases undergo dimerization or specific proteolysis, they become active and, through a cascade of proteolytic reactions, initiate all the biochemical and morphological changes that constitute the apoptosis process [287,300].

The increase in apoptosis caused by PH is supported by studies indicating that prenatal hypoxia enhances death signaling through increased activity of caspase 3 and Fas mRNA while suppressing survival pathways through reduced expression of Bcl-2 and Hsp70 in the hearts of the fetus [306]. The reduction in the bioavailability of NO and the increase in its cytotoxic forms enhance the sensitivity of cells to signals transmitted through Fas receptors. Studies demonstrate the role of peroxynitrite in triggering the process of apoptotic cell death in the context of decreased CuZn-SOD levels [307]. Our recent studies have confirmed that modeling PH leads to a significant reduction in the expression of CuZn-SOD in the cytosol of the heart in 1- and 2-month-old offspring against the backdrop of increased nitrotyrosine levels and a disproportionate expression of iNOS/eNOS [253].

The extrinsic mechanism of apoptosis begins with the binding of specific ligands, known as “death ligands” (such as FasL), to specific transmembrane receptors, such as Fas/CD95/Apo1, or the binding of tumor necrosis factor alpha (TNF-α) to its receptor. In endothelial cells after hypoxia, H_2_O_2_ stimulates the activity of iNOS, which contributes to oxidative cell damage and apoptosis [149,308,309]. NO and its cytotoxic forms lead to the opening of the mitochondrial permeability transition pore (mPTP), which results in the release of cytochrome c and the initiation of the caspase cascade of apoptosis [310,311].

NO and its derivatives can cause peroxidative oxidation of phospholipids and oxidation of thiol groups in mitochondrial membrane proteins, which also leads to the release of apoptotic factors into the cytosol [115,312,313]. Data have also been obtained confirming the direct action of excess NO on the induction of apoptosis in a cGMP-dependent manner in isolated cardiomyocytes [314]. In fetuses subjected to PH, a decrease in the expression of Bcl-2 mRNA, an anti-apoptotic protein, was observed in the left ventricle [315]. Physiological concentrations of NO, produced in vivo by eNOS, can activate cGMP-dependent protein kinases, which in turn influence the proteins involved in apoptotic cascades (such as Bcl-2). A reduction in eNOS activity and NO deficiency may impact the expression of Bcl-2. Bcl-2 can inhibit apoptosis mediated by NO and its derivatives, as well as the cleavage of poly(ADP-ribose) polymerase [316].

Excess NO due to iNOS activity neutralizes the anti-apoptotic members of the BCL-2 family by activating the ASK1-JNK1 pathway, leading to BAX/BAK-dependent cell death. NO and its cytotoxic forms can cause direct S-nitrosylation of cysteine residues in thioredoxin, thereby releasing ASK1 to induce cell death. The mechanism through which NO activates the ASK1-JNK1 axis to initiate BAX/BAK-dependent cell death involves the generation of reactive oxygen species (ROS) and is associated with the formation of peroxynitrite [317,318,319].

## 4. Molecular Mechanisms and Stress Responses

### 4.1. The Interaction Between NO and HIFs in the Myocardium After PH

HIFs function as transcriptional regulators, controlling the expression of genes responsible for the adaptive physiological responses to hypoxic or ischemic conditions [320]. Under low-oxygen conditions, HIFs display cytoprotective effects by promoting tissue repair mechanisms and enhancing the expression of free radical scavengers, including heme oxygenase-1, VEGF, and angiopoietin [267,268]. During hypoxia, HIF-1 modulates cellular energy metabolism by activating alternative ATP-generating pathways, upregulating glutathione production, and strengthening cellular defenses against oxidative stress [321].

A sufficient amount of research has shown that the concentration of HIF-1 and its forms varies in different types of hypoxia, its duration, and across various organs [322,323,324,325,326]. Under conditions of increased nitrosative stress and elevated levels of cytotoxic NO products, along with ATP depletion in tissues, a decrease in HIFs is observed, associated with the activation of the ubiquitin-independent degradation pathway of oxidatively modified HIF-1α and the suppression of its synthesis during ATP deficiency [254,327,328].

The regulation of HIF activity involves intricate control mechanisms affecting the HIFα subunit through both oxygen-dependent and oxygen-independent pathways. Oxygen-independent mechanisms include the regulation of HIFA gene expression by transcription factors such as nuclear factor kappa B (NF-κB), specificity protein 1 (SP1), and NF-E2-related factor 2 (NRF2), which can, in turn, be regulated by active ROS, cytokines, and/or lipopolysaccharide (LPS)-dependent signaling through pathways involving protein kinase C (PKC), inhibitor of NF-κB kinase (IKK), and/or phosphoinositide 3-kinase (PI3K) [329,330,331]. The levels of HIFA mRNA and/or translation can be regulated by microRNAs (miRNA), long non-coding RNAs (lncRNA), and/or angiotensin II-mediated signaling involving PI3K. Post-translational modifications, such as phosphorylation, sumoylation (SUMO), acetylation (Ac), and NO-mediated S-nitrosylation, further contribute to the oxygen-independent regulation of HIFα stability and activity [332,333]. Our findings indicate that PH leads to downregulation of HIF-1 mRNA expression in the hearts of 1- and 2-month-old animals, which coincides with decreased expression of eNOS and reduced levels of NO metabolites. In this same study, we also found that the suppression of HIF-1 mRNA expression after intrauterine hypoxia occurs in the context of HSP70 deficiency [254].

It is known that HSP70 prolongs the “lifespan” of HIF-1 under hypoxic conditions [334,335]. The reduction in HIF-1 mRNA expression in the hearts of rats after prolonged perinatal hypoxia has also been confirmed by other authors [265,266]. And, in our opinion, this may indicate the exhaustion of compensatory-adaptive responses. The regulatory role of NO in the regulation of HIF-1α mRNA expression is well known. Physiological concentrations of NO caused a faster but temporary accumulation of HIF-1α compared to higher doses of the same NO donor. Cytotoxic forms of NO suppressed the expression of HIF-1α. The regulation of HIF-1α by NO is an additional important mechanism through which NO can modulate cellular responses to hypoxia in mammalian cells. NO not only modulates the HIF-1 response under hypoxic conditions but also functions as an inducer of HIF-1 [336,337].

### 4.2. NO and Inflammation After PH

NO is a signaling molecule that plays a key role in the pathogenesis of inflammation. Under physiological conditions, NO exhibits anti-inflammatory properties, contributing to the maintenance of tissue homeostasis. Conversely, in pathological states, excessive NO production transforms it into a pro-inflammatory mediator, promoting inflammatory responses. NO is synthesized and released by endothelial cells via nitric oxide synthases (NOSs), which convert arginine into citrulline, producing NO in the process. Oxygen and NADPH are essential cofactors in this conversion. NO is believed to cause vasodilation in the cardiovascular system, and in addition, it plays a role in the immune responses of activated cells. Moreover, NO is actively involved in the pathogenesis of various inflammatory diseases [11,338].

Cytotoxic forms of NO and ROS in the hypoxia of newborns can trigger the production of oxygen stress-induced high-mobility group box-1 (HMGB-1), an endogenous protein of molecular structures associated with damage (DAMPs), which is linked to toll-like receptor (TLR)-4. This activation leads to the stimulation of nuclear factor kappa B (NF-κB), resulting in the production of an excessive amount of inflammatory mediators. Peroxynitrite, ROS, and inflammatory mediators are produced not only in activated inflammatory cells but also in non-immune cells, such as endothelial cells and cardiomyocytes [339].

Excessive inflammation during hypoxia in newborns and after PH exacerbates tissue/organ damage through the expression of genes and proteins. Elevated cytokines activate various inflammatory immune cells through receptors, including toll-like receptors (TLRs), further increasing the production of cytokines and other inflammatory mediators. For example, the concentration of cytokines such as interleukin (IL)-1β, IL-6, IL-8, and tumor necrosis factor (TNF)-α has increased [340,341,342,343]. Hypoxia-induced IL-1β also increases the expression of the NF-κB gene. Thus, both ROS and hypoxia can activate the NF-κB pathway.

The activated NF-κB pathway, in turn, increases the expression of inflammatory mediator genes and regulates RNS, including the synthesis of iNOS and the production of NO. During inflammation, the increase in HIF-1 induces iNOS and reduces L-arginine, leading to an increase in the production of ONOO and hydroxyl radicals, further reactivating nitrosative stress [344,345,346,347,348].

### 4.3. NO and HSP70 After PH

Recently, a number of studies have focused on the role of heat shock proteins (HSPs) in hypoxia, particularly prenatal hypoxia [349,350,351,352]. HSPs are synthesized in cells of all living organisms in response to various stress factors, including cerebral ischemia. However, the genes for these proteins are activated not only under stress conditions but also during the normal life processes of the cell, including proliferation, differentiation, and apoptosis [353,354,355]. Proteins of this class are involved in all life processes of tissues, organs, and the entire organism.

The most studied protein of this family is the 70 kDa molecular mass protein, HSP70. HSP70 is an inducible representative of the heat shock protein family [356,357,358]. HSP70 is the first protein to be called a chaperone. The function of chaperones in the cell is to bind to damaged or newly synthesized polypeptides and assist them in adopting their native conformation; chaperones also participate in the delivery of proteins to specific organelles [359,360,361]. Chaperones are capable of identifying hydrophobic regions in target polypeptides that are exposed to damaged proteins or may open up in normal, mature cellular proteins during conformational changes. Such conformational changes occur, for example, as a result of cascade modifications of proteins during the process of cellular signal transduction [362,363,364].

Proteins from the HSP70 family are some of the main components of the cellular protein quality control system. Chaperone activity is generally associated with the protective function of HSP70 [365,366,367]. The fact that the chaperone protects cells from a wide range of factors, including those inducing apoptosis, in an ATP-mediated manner and removes irreparable proteins through the proteasomal machinery has been confirmed in numerous in vitro and in vivo experiments using a variety of experimental models. Additionally, much evidence supports the protective effect of HSP70 [115,368].

The results of experiments have highlighted several directions for the practical use of the protective properties of HSP70. First, the organism’s resistance to stress conditions can be increased by enhancing the intracellular content of HSP70 [369,370,371]. Secondly, a number of studies show interest in utilizing the protective properties of extracellular HSP70. Once outside the cell, HSP70 likely interacts with neighboring cells and protects them from cell death. Thus, exogenous HSP70 has demonstrated protective properties similar to those of the intracellular chaperone [372,373,374,375].

Recently, there have been data regarding the regulatory effect of heat shock proteins on mitochondrial dysfunction, which develops during brain ischemia, myocardial ischemia, and prenatal hypoxia, as a result of the pathobiochemical cascade of events [335,376,377,378].

Thus, it would be reasonable to assume that HSP70 is involved in the regulation of signaling pathways in the cell’s response to hypoxia at the level of HIF-1α regulation [334,379,380]. The cytoprotective effect of HSP70 under hypoxic conditions is realized through its anti-apoptotic and mitochondria-protective activities. It is well known that depending on the concentration of ROS, oxidative stress ultimately leads to either necrosis or apoptosis [335,381,382]. A high level of ROS causes significant damage to proteins, lipids, and nucleic acids, leading to necrosis. Moderate oxidative stress results in programmed cell death—apoptosis. HSP70 and HIF-1α, through their prolonged action on the synthesis of antioxidant enzymes, chaperone activity, and stabilization of active filaments, prevent the development of necrosis [383,384].

A number of authors indicate that an increase in HSP70 levels leads to the normalization of the glutathione linkage in the thiol-disulfide system and enhances the resistance of cells to ischemia [115,335,385,386]. The introduction of exogenous HSP70 leads to an increase in the functional activity of the glutathione system [351,387,388]. In other words, it has been shown that HSP70, proteins with pronounced cytoprotective properties, under hypoxic conditions, mobilize antioxidant resources, particularly increasing the levels of both cytosolic and mitochondrial glutathione, which prevents the development of oxidative stress [381,389].

Moreover, it is known that by modulating the level of endogenous reduced glutathione, the expression of heat shock proteins can be regulated within the cell [351]. The deficit of HSP_70_ in the neuron, in the case of reduced glutathione, is, in our opinion, associated with the hyperproduction of ROS and cytotoxic forms of nitric oxide, which lead not only to the modification (reversible and irreversible) of macromolecules, including HSP_70_ itself, but also to a decrease in the expression activity of genes encoding its synthesis. By stabilizing oxidatively damaged macromolecules, HSP_70_ is able to prevent the opening of the mitochondrial pore, thereby blocking the release of cytochrome C from the mitochondria, thus exerting a direct anti-apoptotic effect [11,390,391].

HSP70 plays an important role in preventing oxidative stress. However, a sudden depletion of endogenous glutathione can reduce the expression of HSP70 in tissues/organs under hypoxia, which leads to increased oxidative damage to macromolecules and an enhancement of hypoxic changes [392,393,394,395]. It has been shown that under in vitro conditions, HSP70 is capable of preventing the aggregation of oxidatively damaged citrate synthase, glutathione-S-transferase, glutathione reductase, superoxide dismutase, lactate dehydrogenase, malate dehydrogenase, and regulating the thiol-disulfide balance [351,352].

Moreover, one of the main functions of HSP70 is the induction and prolongation of the stable form of HIF-1α, which triggers further adaptive responses in the cell. We have established that HSP70 “prolongs” the activity of HIF-1α and also independently maintains the expression of NAD-MDH-MH, thereby sustaining the activity of the compensatory ATP production mechanism—the malate–aspartate shuttle mechanism—for an extended period [396].

Thus, it can be concluded that HSP70 is an inevitable companion of the pathobiochemical reactions that develop as a result of PH. The current level of knowledge of the pathophysiological and pathobiochemical processes occurring during hypoxia (ischemia) allows for the consideration of pathogenetic correction of metabolic and morphofunctional changes using pharmaceuticals, for which HSP70 will be the target of action.

We have established that modeling PH leads to a deficiency of HSP70 in the 1- and 2-month-old offspring against the background of glutathione deficiency and the expression of glutathione-dependent enzymes (GPX1 and GPX4) [253,254]. We have shown that significant activation of nitrosative stress leads to a decrease in HSP70 concentration. The modeling of nitrosative stress in vitro was conducted by introducing a dinitrosyl iron complex (DNIC) into the neuron suspension at a cytotoxic concentration of 250 µmol. DNIC is an unstable complex of divalent iron, nitric oxide, and ligands [350,397].

Being a stronger nitrating agent, DNIC interacts with -SH groups, glutathione, cysteine residues of proteins, enzymes, transcription factors, and DNA, forming S-nitrosothiols and N-nitrosothiols [398]. After existing for several minutes, DNIC decomposes, releasing a large amount of nitric oxide and free iron. Free iron catalyzes the Haber–Weiss reaction, leading to the formation of hydroxyl radicals, which are capable of oxidizing proteins, degrading cellular membrane lipids, and damaging DNA. In turn, a large amount of nitric oxide released from the complex exhibits its toxic properties in the form of dinitrogen trioxide (N_2_O_3_) and peroxynitrite (ONOO^−^), which, when excessively synthesized in vivo, lead the organism into a state of nitrosative and oxidative stress [399,400,401].

At the 60th minute of observation, the HSP70 protein content decreased by 51.4% (*p* < 0.05) compared to the intact group. The results of our research show that the decrease in GSH levels at the 60th minute of observation was accompanied by a low level of HSP70 protein, which is confirmed by the strong correlation between GSH and HSP70 (Pearson’s multiple correlation coefficient (R = 0.94678)). We also established a strong negative correlation between HSP70 and the marker of nitrosative stress, nitrotyrosine (Pearson’s multiple correlation coefficient (R = −0.8899)) [402].

It can be assumed that intrauterine hypoxia leads to a decrease in HSP70, which plays a role as an endogenous cytoprotective factor through nitrosative stress in response to the disruption of the nitroxidative system.

### 4.4. Oxidative Stress in Myocardial Damage After PH

Modeling PH leads to the development of postnatal heart dysfunction. In our previous study, it was established that the use of this PH model results in a decrease in myocardial contractile activity and dysfunction of the sinus node. In the contractile myocardium and conduction system, cells with signs of apoptosis and dystrophy are observed, with a certain correlation between the severity of morphological changes and bioelectrical disturbances in rhythm and conduction [79]. The final result of hypoxic heart injury is focal dystrophy [23]. Molecular analyses confirmed this, showing an elevated concentration of ST2 in the blood of animals subjected to PH. ST2 serves as a highly sensitive indicator of myocardial remodeling processes and an increased risk of heart failure [403].

Our findings regarding elevated nitrotyrosine levels after PH align with previous research showing increased cardiac oxidative stress markers in both male and female rats subjected to intrauterine hypoxia [149,404]. Increased oxidative stress is closely associated with cardiovascular diseases such as hypertension and coronary artery disease. It leads to hypertrophy, fibrosis, and apoptosis, which result in impaired heart function [405,406,407].

The characteristic feature of myocardial damage after PH, according to many authors, is the prenatal damage to myocardial mitochondria caused by hypoxia. This makes the mitochondria a source of reactive oxygen species and pro-apoptotic proteins. Against the background of impaired energy production (decreased ATP), this leads to significant activation of oxidative stress and apoptosis [171,172,408,409,410].

According to findings from several studies, intrauterine hypoxia in rats leads to elevated levels of mitochondrial cytochrome C in the bloodstream, accompanied by a decline in both the average mitochondrial density and cristae density, along with decreased expression of mitochondrial manganese superoxide dismutase (Mn-SOD) [411,412,413]. It has also been shown that intrauterine hypoxia modifies the expression of 48 genes involved in metabolic and oxidative stress responses, including genes encoding glutathione-S-transferase subunits and cytochrome C oxidase [203,414,415,416,417].

The impact of intrauterine hypoxia on eNOS expression is duration-dependent, with prolonged ROS overproduction contributing to reduced NO bioavailability and downregulation of eNOS expression [23,68]. The balance between ROS and NO, in the context of sufficient endogenous reduced thiol compounds, determines vascular tone. This is because hypoxia-induced increases in ROS, low NO production, and a deficiency of reduced low-molecular thiol compounds in the fetus lead to the formation of cytotoxic derivatives of nitric oxide, which enhances peripheral vasoconstriction and exacerbates myocardial ischemia. An excess of NADPH during prenatal hypoxia is a key factor in the formation of ROS, which can react with NO to form the stable peroxynitrite anion, thereby reducing the bioavailability of NO [417,418,419,420].

## 5. Cardioprotection and Therapeutic Approaches

### 5.1. Cardioprotection After PH

According to several researchers, when modeling hypoxia–ischemia–reoxygenation conditions, the oxygen transport compound—perfluorane emulsion—demonstrated a high degree of pharmacological cardioprotection efficacy. Solutions of adenosine triphosphate (ATP), cocarboxylase, magnesium sulfate, riboxin, solcoseryl, cytochrome C, and essentiale showed a medium degree of efficacy. Low efficacy was observed with ascorbic acid solution and L-carnitine chloride solution (levocarnitine) [421,422,423,424,425,426,427,428,429].

For a long time, the most significant group of substances that could be classified as regulatory antihypoxants consisted of nonspecific activators of enzymatic and coenzyme systems. They were considered the only clinically available drugs of this kind. These include B vitamins, thiol derivatives, and pyrimidine derivatives. However, even those that were once extremely popular, with numerous publications on their successful clinical application, are unlikely to remain in the arsenal of resuscitation specialists, although the theoretical grounds for their implementation in practice were promising [430,431]. For example, the conversion of nicotinamide to NAD requires a series of directed synthesis reactions, which are inhibited by oxygen deficiency, and therefore, the final antihypoxic effect of nicotinamide is small and unstable [432,433].

Recently, antihypoxants have been discovered that do not lower body temperature or oxygen consumption, do not stimulate gluconeogenesis, almost halve glycolysis, and do not possess antioxidant properties. The mechanism of their action is unknown and cannot yet be linked to any of the previously discussed mechanisms, but their protective effect significantly exceeds that of “first” and “second” generation antihypoxants. One such drug is Meldonium (Grindex, Tinley Park, IL, USA), synthesized in the early 1980s at the Institute of Organic Synthesis in Latvia. Structurally, Meldonium is a synthetic analog of the precursor in carnitine biosynthesis—gamma-butyrobetaine. It has been established that, like carnitine, it participates in the energy metabolism of cells [434,435,436,437]. Thus, it prevents the activation of glycolysis reactions, which dominate under tissue hypoxia conditions, and therefore exhibits cytoprotective effects.

The anti-ischemic and cytoprotective effects of Meldonium on the postoperative course after open-heart or brain surgery interventions have been demonstrated in various studies [438,439,440]. Its effects are especially pronounced when its use is started 2–3 days before the surgery and continued afterward. With a single dose, the drug had an optimizing effect on the cerebral circulation system when there were significant impairments in the reactivity of the cerebral blood vessels. At the same time, with unstable hemodynamics after a single intravenous administration of Meldonium, a clear increase in systemic blood pressure was observed, and the quality characteristics of circulation improved [441,442].

It has been shown that the prophylactic administration of succinic acid, aminothiol antihypoxants such as gutimine and amtyzol, as well as succinate-containing aminothiol antihypoxants like gutimine succinate and amtyzol succinate, exhibits pronounced antihypoxic and antioxidant effects [443]. There is an antihypoxant called Lipin. Lipin, a modified egg phosphatidylcholine (lecithin), exhibits antihypoxic effects, promotes the increased diffusion rate of oxygen from the lungs into the blood and from the blood into tissues, normalizes tissue respiration processes, restores the functional activity of endothelial cells, and stimulates the synthesis and secretion of endothelial relaxation factor. It also improves microcirculation and the rheological properties of blood. Lipin inhibits lipid peroxidation processes in the blood and tissues, supports the activity of the body’s antioxidant systems, exhibits a membrane-protective effect, functions as a nonspecific detoxifying agent, and enhances nonspecific immunity. When administered via inhalation, it has a positive effect on lung surfactant, improves pulmonary and alveolar ventilation, and increases the rate of oxygen transport through biological membranes [444,445,446].

Thus, in the hearts of newborn rats under conditions of intrauterine hypoxia, reversible changes develop, and in severe cases, irreversible changes occur both in cardiomyocytes (conducting and contractile) and in the vessels of the hemomicrocirculatory bed. This is a manifestation not only of hypoxic damage to the heart muscle but also evidence of an ischemic nature of the myocardial damage that has developed. Therefore, studying the pathways for correcting antenatal hypoxia remains a relevant issue, within which it is advisable to study the effects of modern antihypoxants on the myocardium in experimental settings, as well as the possibilities for their combination to enhance the effect.

### 5.2. NO Modulators—Promising Cardioprotectors After PH

The literature data regarding the role of the NO system in the development of cardiovascular pathology in newborns and the potential cardioprotective effects of its modulators is quite limited. Several studies have established the cardio- and endothelial-protective properties of drugs that are capable of both increasing the synthesis of NO and enhancing the bioavailability of this messenger [114,253,447,448]. In this regard, substances such as L-arginine, Thiotriazoline, Angiolin, and Meldonium are of interest.

L-arginine is a substrate for the formation of NO in endothelial cells of blood vessels, exhibiting antioxidant, cytoprotective, antihypoxic, and membrane-stabilizing properties [449]. Thiotriazoline (morpholinium 3-methyl-1,2,4-triazolyl-5-thioacetate) is a specific scavenger of NO and its cytotoxic forms, enhancing the bioavailability of NO by protecting it from ROS. In ischemic conditions, Thiotriazoline strengthens the compensatory activation of the malate–aspartate shuttle mechanism, reduces the inhibition of oxidation processes in the Krebs cycle while preserving intracellular ATP levels, and demonstrates hepatoprotective, cardioprotective, anti-ischemic, and antioxidant properties [450].

Angiolin ([S]-2,6-diaminohexane acid 3-methyl-1,2,4-triazolyl-5-thioacetate) increases the expression of VEGF and the density of proliferating endothelial cells in muscular-type blood vessels and the microcirculatory bed. It enhances the bioavailability of NO, preserves the ultrastructure of mitochondria during ischemia, and also boosts ATP production in ischemic conditions by activating the compensatory malate–aspartate shuttle mechanism. Angiolin increases the expression of eNOS and exhibits endothelial, cardioprotective, neuroprotective, antioxidant, and anti-ischemic properties [351].

Meldonium reduces the formation of carnitine from its precursor, gamma-butyrobetaine, and the accumulation of the latter stimulates NO synthesis. Meldonium decreases carnitine-mediated transport of long-chain fatty acids across mitochondrial membranes without affecting the metabolism of short-chain fatty acids and activates an alternative energy production system—glucose oxidation [451]. Meldonium exhibits anti-ischemic and cardioprotective properties.

In our studies using the PH model, the cardioprotective effects of NO modulators—L-arginine, Thiotriazoline, Angiolin, and Meldonium—were demonstrated for the first time, with varying degrees of expression. The presented drugs reduced the concentration of ST2 protein, normalized the expression of iNOS mRNA and eNOS mRNA, as well as iNOS and eNOS proteins, increased the concentration of HSP70 and HIF-1, and decreased the marker of nitrosative stress—nitrotyrosine—in the blood and myocardium of 1- and 2-month-old offspring.

Two drugs—Angiolin and Thiotriazoline—were able to have a full impact on endothelial dysfunction indicators after PH (reducing sEPCR with increased Tie-2, VEGF-B, and Cu/ZnSOD, GPX), which perform protective and antioxidant functions [144,253]. The cardioprotective effect of NO modulators was also manifested in the improvement of the electrophysiological properties of the heart in the offspring after PH. The most pronounced therapeutic effect was observed with Angiolin and Thiotriazoline, which contributed to almost complete normalization of heart rate, while Angiolin also restored the neurogenic control of the sinus node’s automaticity. The obtained results allowed us to rank the therapeutic efficacy of the used drugs in descending order: Angiolin > Thiotriazoline > Meldonium > L-arginine in eliminating disturbances in the electrical activity of the heart [79].

#### 5.2.1. Angiolin((S)-2,6-Diaminohexanoic Acid 3-methyl-1,2,4-triazolyl-5-thioacetate)

Angiolin exhibits pronounced endothelial-protective, neuroprotective, cardioprotective, energotropic, antioxidant, anti-ischemic, and anti-inflammatory properties. Angiolin acts as an anti-ischemic and antioxidant agent with a significant effect on the endothelium of the brain and heart vessels and metabolism. Its neuroprotective properties are due to the conversion of L-lysine into pipecolic acid, which enhances the affinity of the GABA-benzodiazepine receptor complex, thereby reducing manifestations of glutamate excitotoxicity. The drug significantly reduces neuronal death in ischemic and hemorrhagic strokes, normalizes the functioning of the compensatory GABA-shunt, and increases the ATP level in brain tissues. Under acute ischemic conditions in the brain, the drug preserves the functional activity of neuronal mitochondria [452,453].

The drug exhibits pronounced antioxidant properties, activates the glutathione branch of the thiol-disulfide system, increases the activity of glutathione peroxidase and glutathione transferase, reduces the formation of reactive oxygen species, and decreases the accumulation of markers of oxidative and nitrosative stress. The endothelial-protective effect of the drug in cerebrovascular disorders and hypertension is due to its ability to regulate NO production, reduce the formation of peroxynitrite and homocysteine, increase the activity of superoxide dismutase and NO-synthase, and preserve reduced thiol groups and L-arginine. The drug increases the bioavailability of NO and can improve its transport to target cells when endothelial function is impaired in brain blood vessels.

In cerebrovascular disorders and vascular surgeries, the drug preserves the morphofunctional parameters of endothelial cells in muscle-type vessels and capillaries of the brain, increases RNA content in the nuclei and cytoplasm of endothelial cells, activates their proliferation, and increases the binding coefficient of vascular endothelial growth factor (VEGF) with the endothelium. The drug demonstrates anti-inflammatory properties, reducing the expression of the pro-inflammatory cytokine IL-1β [452,453,454].

The cardioprotective properties of the drug are aimed at increasing the survival rate of cardiomyocytes during acute myocardial ischemia, improving ECG parameters. The drug improves overall and cardiodynamics during acute myocardial ischemia—normalizing systolic blood pressure, reducing ischemic left ventricular dysfunction, increasing left ventricular pressure, and increasing working and systolic heart indices while lowering overall peripheral vascular resistance. In angina and myocardial infarction, Angiolin improves myocardial energy metabolism by intensifying aerobic ATP formation reactions and activates the compensatory malate–aspartate shuttle for ATP production [351,452,453,454]. When administered parenterally, the average time for Angiolin’s presence in the body is 20 min. The distribution coefficient in the body is 464 µg/g/min, indicating a rapid reach of the drug to target organs and tissues. A total of 17% of the administered dose of the drug binds to plasma proteins. Within the first 3–5 min, 34.7% of the administered dose of Angiolin reaches the myocardium. The time to reach the maximum concentration in the myocardium is 7.8 min. The maximum concentration in the myocardium (Cmax cor) is 16.7 µg/g. The clearance rate from the myocardium for Angiolin is 324 µg/min. The drug is excreted via the kidneys, with 62% of the dose eliminated in unchanged form, and its renal clearance is 3.6 mg/min [455].

According to acute toxicity studies of the drug “ANGIOLIN” (mice, rats, rabbits), it was classified as a Class V of toxicity (practically non-toxic substances with no cumulative properties according to the accumulation index). The drug “ANGIOLIN” does not cause skin irritation on undamaged rat skin, does not have a local irritant effect on the undamaged conjunctiva of guinea pigs’ eyes, does not cause allergic reactions in guinea pigs, and does not have ulcerogenic effects in rats. In a 180-day intragastric administration study of “ANGIOLIN” in doses of 100, 500, and 1000 mg/kg, it was found that the drug does not cause structural–functional changes, does not lead to dystrophic or hemodynamic disorders, nor does it provoke destructive reactions in the studied tissues of animals [455].

The investigated drug “ANGIOLIN” has a good safety profile, has passed the first phase of clinical trials, and, by approval from the State Expert Center of the Ministry of Health of Ukraine, has been authorized for the second phase of clinical trials. The primary cardioprotective action of Angiolin, when administered after intrauterine hypoxia, with sustained effects even after a one-month cessation, can be explained by the following properties.

In conditions of acute brain ischemia, Angiolin exhibits pronounced endothelial-protective properties—it maintains endothelial cell density, increases RNA concentration in cell nuclei, boosts the density of proliferating endothelial cells (BrdU test), enhances the utilization of endogenous L-arginine, and increases the expression of vascular endothelial growth factor (VEGF) and eNOS. Additionally, the presence of divalent sulfur in its structure gives it the property of scavenging NO. Angiolin, together with vitamin C, forms L-carnitine and normalizes mitochondrial function.

Our studies have shown that Angiolin improves the ultrastructure of neurons in the CA1 zone of the hippocampus in chronic cerebral ischemia (reducing crystal destruction, uneven electron density of the matrix, and increasing mitochondrial density). It also lowers intra-mitochondrial iNOS levels and raises the concentration of cytosolic and intra-mitochondrial HSP70. Our work has demonstrated that Angiolin can activate the malate–aspartate shuttle mechanism in the myocardium during ischemia. The positive effect on the NO system and the reduction of oxidative stress, along with the increase in mRNA of HIF-1 and HSP70, seem to provide Angiolin with its cardioprotective effect after intrauterine hypoxia.

It is noteworthy that the cardioprotective effect of Angiolin in offspring after PH was sustained even one month after the discontinuation of the drug. Experimental studies have shown that Angiolin increases the expression of eNOS mRNA and eNOS activity in ischemic myocardium in rats. Angiolin also enhances the expression of VEGF and the binding coefficient of VEGF to endothelial cells, as well as the density of endothelial cells and proliferating endothelial cells in the capillary network and vessel walls. It increases RNA concentration in endothelial cells under hypoxic and circulatory ischemic conditions [456].

Angiolin exerts a protective effect on nitric oxide (NO) and enhances its bioavailability. Due to its inherent instability and short half-life, NO extends its activity by forming stable S-nitrosylated complexes with low-molecular-weight compounds such as glutathione and cysteine. A deficiency of such compounds leads to a marked reduction in NO bioavailability. When low-molecular thiols are deficient, NO, under the influence of ROS, transforms into peroxynitrite, which can trigger the initiation of nitrosative stress [457]. Angiolin, due to its chemical structure, acts as a spin trap and can form a complex with NO. Angiolin positively affects the state of the nitric oxide system in the myocardium under experimental ischemia—it increases NO synthesis by enhancing eNOS expression, increases NO bioavailability, and reduces parasitic reactions by decreasing iNOS hyperactivity. The mechanism of its effect on eNOS expression can be explained by Angiolin’s influence on HSP70 and HIF-1α. Angiolin prolongs the “lifetime” of HIF-1α through HSP70 mechanisms. Angiolin also has a positive effect on the glutathione thiol-disulfide system, which is coupled with the NO system. Experimental studies of myocardial ischemia have demonstrated that Angiolin enhances the activity of glutathione reductase and glutathione peroxidase while elevating reduced glutathione levels in the myocardial cytosol of rats [335]. As is well known, HIF-1α increases the expression of eNOS and VEGF during hypoxia and ischemia [458].

Angiolin, due to its positive effect on the NO system, positively influenced cardio- and hemodynamics during experimental myocardial ischemia. Administration of Angiolin to rabbits with occlusion of the descending coronary artery led to the restoration of left ventricular dysfunction, which was expressed in an increase in the left ventricular work index and left ventricular stroke work index, an increase in left ventricular pressure, and a decrease in total peripheral vascular resistance [456]. We have shown that Angiolin can have a significant impact on endothelial dysfunction markers after PH (decrease in sEPCR along with an increase in Tie-2, VEGF-B, Cu/ZnSOD, GPX), which perform protective and antioxidant functions. Angiolin (S)-2,6-diaminohexanoic acid 3-methyl-1,2,4-triazolyl-5-thioacetate, which has scavenger properties for NO, with fragments of its chemical structure participating in this process, can form nitrothiols and enhance the bioavailability of NO. Angiolin normalizes the expression of eNOS/iNOS. Studies on a rat model of cerebral ischemia have demonstrated the endothelial-protective activity of Angiolin, including increased endothelial cell density in muscular-type vessels and the microcirculatory bed, an increase in the density of proliferating endothelial cells, as well as an increase in the expression of vascular endothelial growth factor (VEGF) and its receptor binding coefficient [11,456].

There is data indicating that VEGF enhances the regulation of the enzyme eNOS and induces a biphasic stimulation of endothelial NO production [459]. This suggests the possibility of eNOS expression being mediated through VEGF under the influence of Angiolin. Angiolin may affect the expression of endothelial-specific factors and antioxidant enzymes by influencing the thiol-disulfide system through the enhancement of glutathione levels and regulating post-translational mechanisms. Based on the conducted studies, the experimental justification for further preclinical investigation of Angiolin as a promising cardioprotective agent after PH has been established.

#### 5.2.2. Tiothiazoline (Morpholine 3-methyl-1,2,4-triazolyl-5-thioacetate; Morpholine Thiazotate)

The history of tiothiazoline dates back to the 1960s. Preclinical studies, conducted according to the requirements of the Pharmacological Committee of the Ministry of Health of the USSR and the State Expert Center of the Ministry of Health of Ukraine, have shown that tiothiazoline exhibits high antioxidant, anti-ischemic, cardioprotective, and antihypoxic properties, surpassing reference drugs in terms of strength. It has been established that tiothiazoline has low toxicity when administered through various routes to four types of animals, meaning the drug belongs to Class V of toxicity (practically non-toxic substances). Tiothiazoline does not have cumulative properties, does not exhibit skin-irritating effects on intact skin, does not cause allergic reactions, and does not have ulcerogenic or immunotoxic effects [460].

Toxicological analysis, along with comprehensive behavioral, physiological, and biochemical studies and pathological examination of the internal organs of white rats and dogs, has shown that intraperitoneal, intravenous, and intragastric administration of tiothiazoline at therapeutic, intermediate, and sub-toxic doses over 180 days does not cause structural changes, does not lead to dystrophic and hemodynamic disorders, nor does it cause destructive reactions in the examined tissues of the animals. The administration of the drug does not result in irreversible changes in biochemical markers of liver and kidney functional status. Research has also established that tiothiazoline does not exhibit teratogenic, embryotoxic, mutagenic, or carcinogenic effects [460,461,462]. 

Tiothiazoline exhibits scavenger properties for cytotoxic forms of NO and exerts protective effects on the transport of NO by positively influencing the thiol-disulfide equilibrium and increasing the levels of reduced thiols and glutathione. Furthermore, we hypothesize that tiothiazoline itself may act as a carrier for NO, forming stable S-nitrosyl complexes with it [353,463]. Thiotriazoline is capable of increasing the bioavailability of NO in the presence of excessive ROS. It acts as an antioxidant, a scavenger of ROS and NO, enhancing the activity of glutathione-dependent enzymes and increasing the levels of reduced glutathione during myocardial ischemia. Thiotriazoline (10^−5^–10^−7^ M) in vitro reduced the levels of superoxide radical and peroxynitrite due to the presence of a thiol group in its structure. Thiotriazoline prevents the irreversible inactivation of the transcription factor NF-kB, protecting cysteine residues (Cys 252, Cys 154, and Cys 61) in its DNA-binding domains, which are sensitive to excess ROS. Thiotriazoline may participate in the restoration of these groups during reversible inactivation, acting as a Redox Factor-1. It enhances the activation of the expression of redox-sensitive genes necessary to protect cells from oxidative stress. Additionally, tiothiazoline reduces the intensity of nitrosative stress and increases eNOS (endothelial nitric oxide synthase) activity [351,464,465,466].

Thiotriazoline exhibits cardioprotective effects by positively influencing energy metabolism in the ischemic myocardium. It increases ATP levels during ischemia and hypoxia by normalizing the Krebs cycle, enhancing glucose utilization, and promoting the oxidation of free fatty acids [467]. Thiotriazoline stimulates lactate dehydrogenase, promoting the conversion of lactate to pyruvate. This not only eliminates lactate acidosis and normalizes intracellular pH but also stimulates the Krebs cycle by increasing the pyruvate levels. Additionally, Thiotriazoline demonstrates antioxidant properties. Numerous studies have established its ability to reduce the formation of end products of oxidative and nitrosative stress while enhancing the activity of Cu/ZnSOD, GPX1, and GPX4 in the liver, heart, and brain of animals with various experimental pathologies [353]. It is known that Thiotriazoline also demonstrates a cardioprotective effect and enhances the endurance of animals under working hypoxia by increasing HIF-1 and preserving mitochondrial ultrastructure. Due to its antioxidant action, Thiotriazoline maintains receptor sensitivity thresholds, preserves membrane fluidity, and protects phospholipids from oxidation. Thiotriazoline also enhances the effectiveness of arginine when used in combination. The pharmacological effect of this combination is attributed to the positive impact on the synthesis, transport, and bioavailability of NO, as well as the physiological functions of this molecular messenger [468]. In rats with modeling isadrin–pituitrin myocardial infarction, Thiotriazoline stimulated LDH in the direction of the formation of pyruvate from lactate, which eliminated lactic acidosis and normalized intracellular pH and stimulated the Krebs cycle by increasing pyruvate [469].

In the same experimental mode, Thiotriazoline activated the malate–aspartate shunt in the myocardium in the acute period of myocardial infarction [450]. The study involving 8298 patients with various cardiovascular diseases, including 5700 patients with different forms of coronary artery disease, deserves attention. Proven effects of Thiotriazoline include a significant reduction in the number of ventricular arrhythmias and correction of rhythm disorders (ectopic beats, paroxysmal atrial fibrillation, sinus node dysfunction). The drug also has an application in myocardial dystrophies, as it works under both oxygen deficiency and sufficient oxygen conditions. Furthermore, Thiotriazoline improves the metabolism not only of cardiomyocytes but also of cells in the CNS, liver, etc. Importantly, Thiotriazoline had a positive effect on quality of life, as assessed by classical methods using the Minnesota questionnaire and the Nottingham health profile: improvement in overall quality of life indicators, increased physical activity, and enhanced emotional well-being [470]. Thiotriazoline administration (600 mg/day) to 8298 patients with Class II–III stable angina pectoris reduced the number of weekly angina attacks by 46.32%; in the control group, it was reduced by 33.24% (*p* = 0.028), respectively (*p* = 0.031), and increased exercise tolerance [471]. Our research has revealed that Thiotriazoline exhibits a significant cardioprotective and endothelialotropic effect after PH, which is implemented through the modulation of NO [114,253]. The obtained results justify the potential for conducting additional preclinical and clinical studies of Thiotriazoline (as an approved drug) as a treatment for cardiovascular system pathologies following intrauterine hypoxia.

#### 5.2.3. Mildronate

Mildronate (3-(2,2,2-trimethylhydrazine) propionate) reversibly blocks gamma-butyrobetaine hydroxylase, which catalyzes the conversion of gamma-butyrobetaine into carnitine, thus significantly inhibiting the intake of carnitine, which is responsible for the transport of fatty acids through the membrane into muscle cells. This effect of Mildronate is accompanied by a reduction in carnitine-dependent oxidation of free fatty acids (FFAs) and, as a result, leads to the activation of glucose oxidation, which is more energy-efficient under ischemic conditions [472,473]. Treatment with Mildronate is accompanied by a compensatory increase in the expression of a number of genes in the myocardium that code for enzymes involved in lipid metabolism—lipoprotein lipase, fatty acid translocase, carnitine palmitoyltransferase I, and enzymes involved in triglyceride synthesis [451].

Mildronate is capable of improving the contractile function of the myocardium, enhancing hexokinase activity, and modulating the ATP/ADP/AMP ratio by activating AMP-activated protein kinase (AMPK), which helps restore ATP levels [474]. An important feature of Mildronate’s action, which distinguishes it from other drugs affecting myocardial metabolism, is the absence of accumulation of underoxidized fatty acids within the mitochondria and the increase in nitric oxide (NO) production. This occurs because Mildronate inhibits the hydroxylation of γ-butyrobetaine and increases the intracellular pool of γ-butyrobetaine, which, through esterification, exhibits cholinomimetic properties. The esters of γ-butyrobetaine can activate NOS via acetylcholine receptors on endothelial cells. Mildronate improves exercise tolerance and the quality of life in patients, positively affecting the functional parameters of the heart, including ejection fraction and systolic volume, in cases of myocardial ischemia [441].

Our research has confirmed the antihypoxic activity of Mildronate during PH. However, no significant positive effect of Mildronate on the NO system indicators in the myocardium of animals that underwent PH was observed. It was found that Mildronate reliably increased the mRNA expression of iNOS while significantly reducing the concentration of nitrotyrosine, which more strongly indicates its antioxidant properties [475].

#### 5.2.4. L-Arginine

L-arginine is a substrate for the production of nitric oxide (NO) in the endothelial cells of blood vessels, which is a factor responsible for the dilation of peripheral vessels. The NO produced from arginine reduces the overall peripheral vascular resistance and blood pressure and alleviates oxygen deprivation, particularly in the heart tissues [476]. L-arginine exerts antioxidant effects by participating in the transamination cycle and the elimination of nitrogenous waste products from the body, including ammonia, urea, and uric acid, which are by-products of protein metabolism. The ability to synthesize urea and eliminate protein waste depends on the efficiency of the ornithine–citrulline–L-arginine cycle. However, oxidative stress reduces the clinical effectiveness of L-arginine [477].

L-arginine plays an important role in protein synthesis, increasing muscle mass, improving muscle recovery after physical exertion, accelerating wound healing, eliminating waste, optimizing immune system function, and enhancing the production of growth hormone [478]. L-arginine is a common substrate for both NO (nitric oxide) and polyamines (putrescine, spermidine, and spermine). NO and polyamines play an important role in reproduction, embryogenesis, reducing neonatal mortality, and embryonic angiogenesis. NO regulates gene expression and protein synthesis and facilitates the proliferation, growth, and differentiation of the fetus [479]. Therefore, researchers and clinicians have focused on the use of the NO substrate, L-arginine, to reduce the negative consequences of PH [480].

Our studies have revealed the positive effect of L-arginine on the NO system indicators in the hearts of 1- and 2-month-old rats after PH. However, in terms of effectiveness, L-arginine was less potent than the new molecules—Thiotriazoline and Angiolin. Nevertheless, a certain positive influence of L-arginine on molecular indicators of endothelial dysfunction in the hearts of rats after PH was identified.

The weaker effect of L-arginine compared to Angiolin and Thiotriazoline can be explained in terms of the lifespan of NO under ischemic and hypoxic conditions, which are accompanied by oxidative stress. The “newborn” NO is immediately at risk of being “bitten” by the superoxide radical and converted into the harmful peroxynitrite [11,481]. Only combinations of L-arginine with SH-group donors or antioxidants (cysteine, glutathione, Thiotriazoline, Angiolin) are capable of enhancing its NO-modulating activity [351]. The combined action of L-arginine and Thiotriazoline is aimed at the synthesis, stabilization, and enhancement of the bioavailability of NO [335,482,483]. The combined action of L-arginine and Thiotriazoline can be used to correct disorders caused by NO deficiency.

#### 5.2.5. Repurposing Pharmacological Agents for Cardiovascular Protection in Prenatal Hypoxia, Comorbid Conditions, and Long-Term Consequences

Contemporary pharmacological approaches are investigating the repurposing of established drugs to address cardiovascular complications arising from PH, which impact both neonates and adults, with long-term sequelae of hypoxia persisting into adulthood [484,485]. Widely used in the management of type 2 diabetes, metformin exhibits cardioprotective properties through the activation of AMP-activated protein kinase (AMPK) [486,487,488], stimulation of endothelial nitric oxide synthase, and reduction of mitochondrial oxidative stress [489,490,491], which may be beneficial in offspring exposed to PH [492,493]. Antiviral agents developed for the treatment of COVID-19 have shown potential in attenuating systemic inflammation and stress responses [494], which is particularly relevant for individuals with comorbidities [495,496,497]. Emerging evidence suggests that prenatal hypoxia and viral exposure may interact with genetically regulated pathways—including those associated with interferon signaling, nitric oxide metabolism, and hypoxia-inducible factor-mediated mechanisms—thereby amplifying the risk of cardiovascular dysfunction in the offspring [498,499,500]. A potent antioxidant, alpha-lipoic acid can enhance endothelial function, decrease reactive oxygen species levels, and restore cellular redox balance, making it a promising candidate for managing PH-induced nitrosative dysfunction [501,502]. Additionally, natural compounds such as melatonin, resveratrol, and curcumin have demonstrated the ability to modulate NO and hypoxia-inducible factor signaling pathways, thereby reducing inflammation and cardiomyocyte apoptosis [503,504,505]. In patients with comorbid conditions, the inclusion of such agents may offer therapeutic advantages due to their multimodal mechanisms of action [506,507]. Further preclinical and clinical studies are required to comprehensively assess the efficacy, safety, optimal treatment regimens, and long-term outcomes of these repurposed agents in the context of prenatal hypoxia, associated comorbidities, and long-term consequences.

## 6. Conclusions

Therefore, the development and investigation of pharmacotherapeutic strategies targeting the nitric oxide (NO) system for the treatment of prenatal myocardial damage remain a pressing issue in contemporary pharmacology. This provides a theoretical basis for the potential of studying NO system modulators with diverse mechanisms of action—such as L-arginine, Thiotriazoline, Angiolin, and Mildronate—as cardioprotective agents for managing post-hypoxic cardiovascular disturbances in newborns.

## Figures and Tables

**Figure 1 antioxidants-14-00743-f001:**
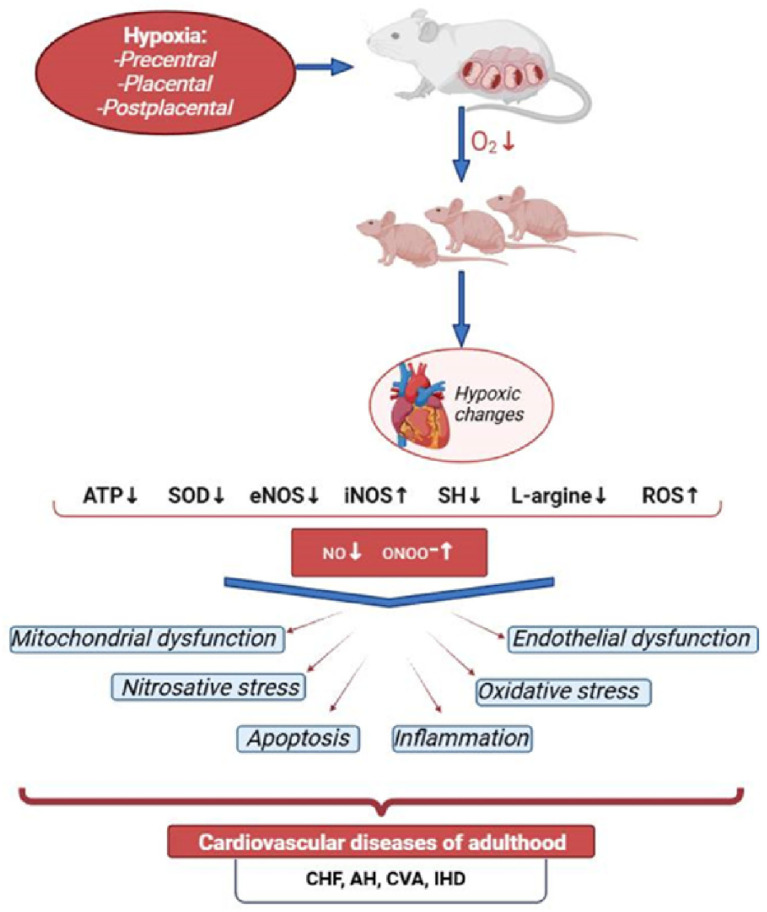
Prenatal hypoxia and its effects on the cardiovascular system of the fetus and offspring. PH causes disruption in the nitric oxide (NO) system, leading to NO deficiency, increased reactive oxygen species (ROS) and their cytotoxic forms, and the activation of NO-dependent molecular–biochemical mechanisms that lead to apoptosis, endothelial dysfunction, mitochondrial dysfunction, inflammation, oxidative stress, and nitrosative stress. This initiates the foundation for the development of cardiovascular diseases in adulthood (chronic heart failure (CHF), arterial hypertension (AH), cerebrovascular accident (CVA), ischemic heart disease (IHD)). Arrows indicate increase and decrease.

**Figure 2 antioxidants-14-00743-f002:**
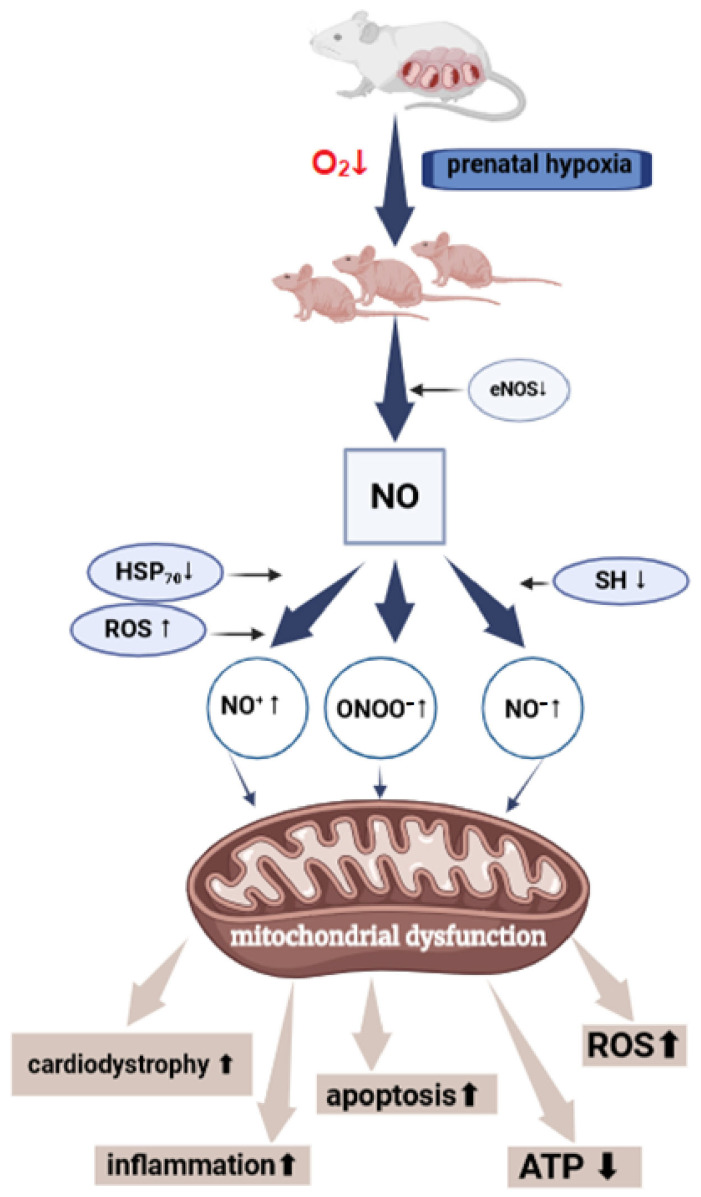
NO-dependent mechanisms of mitochondrial dysfunction formation after prenatal hypoxia. Arrows indicate increase and decrease.

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
