# Peer review of "Molecular and Biochemical Mechanisms of Cardiomyopathy Development Following Prenatal Hypoxia—Focus on the NO System"

_antioxidants, 2025, doi:10.3390/antiox14060743_

Round 1

Reviewer 1 Report

Dear authors of the antioxidants-36000688 paper, I have some constructive comments about your work:
*I find the work itself redundant and the text excessively long.
*The bibliographic citations are presented with deficiencies that impede fluent reading.
*Unfortunately, they omit the cellular apoptotic processes that occur during normal heart development.
*I strongly recommend that you use the International Embryological Terminology to correctly describe the classification of "child," "neonate," etc.

Antioxidants-3600688

Dear authors of the work antioxidents-3600688, I have several constructive comments about your work.

Lines 42-45: The idea requires a bibliographic citation.

Lines 47-50: The idea requires a bibliographic citation.

Line 51: What do you mean by structural changes? Anatomical-morphological changes?

Lines 83-85: The idea requires a bibliographic citation.

Lines 100-103: The idea requires a bibliographic citation.

Line 114: I suggest changing the qualifier: "child." Please use the International Embryological Terminology.

Line 221: I suggest changing the qualifier: "child." Please use the International Embriological Terminology.

Lines 121-132: Ideas require bibliographic citations

Lines 135: Patients? Which patients?

Line 140: I suggest changing the qualifier to "child." Please use the International Embriology Terminology.

Lines 140-146: Statements require bibliographic citations

Lines 157-166: the statement requires bibliographic citation.

Lines 161-166: The marked text is out of context, unless it is what they refer to.

Lines 187-190: The statements require bibliographic citations.

Lines 213-215: The idea requires a bibliographic citation.

Lines 216-217: The idea requires a bibliographic citation.

Lines 220-226: Please redistribute the bibliographic citations (106-109) as appropriate to the ideas presented.

Lines 239-245: Please redistribute the bibliographic citations (120-122) as appropriate to the ideas presented.

Lines 248-251: Please redistribute the bibliographic citations (124-126) as appropriate to the ideas presented.

Lines 257-262: Please redistribute the bibliographic citations (130-132) as appropriate to the ideas presented.

Lines 324-326 The statements require bibliographic citations.

Lines 350-355 Please redistribute the bibliographic citations (115, 166-169) as appropriate to the ideas presented.

Lines 362-364 The statement requires bibliographic citations.

Lines 402-410 The statements require bibliographic citations.

Lines 412-419 Please redistribute the bibliographic citations (79, 172, 188-191) as appropriate to the ideas presented.

Lines 430-441 Please redistribute the bibliographic citations (141, 202-204) as appropriate to the ideas presented.

Lines 442-450 Please redistribute the bibliographic citations (205-207) as appropriate to the ideas presented.

Lines 460-473 Please redistribute the citations. Bibliographical citations (11, 217-221) as appropriate to the ideas presented

Lines 485-494 Please redistribute the bibliographical citations (230-235) as appropriate to the ideas presented

Lines 505-515 Please redistribute the bibliographical citations (230-235) as appropriate to the ideas presented

Lines 516-527 Please redistribute the bibliographical citations (246-251) as appropriate to the ideas presented

Lines 528-540 The statements require bibliographical citations

Lines 571-573 The statement requires a bibliographical citation

Lines 576-581 Please redistribute the bibliographical citations (60, 259-261) as appropriate to the ideas presented

Lines 624-627 The statements require bibliographical citations

Lines 628-630 say a “sufficient number of articles,” however, they only cite one work (271)

Lines 711-715: the statements require bibliographic citations

Lines 736-743: the statements require bibliographic citations

Line 789: the abbreviation had already been described

Lines 803-809: the statements require bibliographic citations

Lines 831-838: several statements require bibliographic citations

Lines 879-885: please redistribute the bibliographic citations (115, 370-373) as appropriate to the ideas presented

Lines 1019-1028: please redistribute the bibliographic citations (435-438) as appropriate to the ideas presented

Lines 1101-1107: the statements require bibliographic citations

Lines 1120-1140: the statements require bibliographic citations

Lines 1141-1146 The statements require bibliographic citations

Lines 1155-1163 The statements require bibliographic citations

Lines 1164-1169 Please include the bibliographic citations for "According to acute toxicity studies"

Lines 1187-1195 The idea requires bibliographic citation

Lines 1249-1258 The statements require bibliographic citations

Lines 1328-1338 Please redistribute the bibliographic citations (456, 477-478) as appropriate to the ideas presented

Author Response

We thank the reviewer for the evaluation of the manuscript and the detailed feedback.

This analytical review paper is prepared to share our perspective on cardiovascular system disorders following prenatal hypoxia and to propose directions for cardioprotection, taking into account current scientific views and our own experience. The article turned out to be quite extensive. However, we expect that it will be of interest to specialists in the fields of fundamental medicine, pharmacology, cardiology, pediatrics, and neonatology.

The authors have taken the suggestions into account and added material regarding apoptotic processes involved in normal heart development. The authors have also made corrections to the formatting of references and included appropriate citations. Furthermore, the comments regarding terminology were also addressed in the revised manuscript.

Reviewer 2 Report

  1. Elevation of Reactive nitrogen species (RNS) is always associated with reactive oxygen species (ROS). There must by be some cross talk between RNS and ROS. It is important to discuss the cross talk between RNS and ROS in the review article.
  2. What about the role of HIF inhibitors in modulating prenatal hypoxia
  3. Nothing is described about prenatal hypoxia and its causes. It is important to add a paragraph about prenatal hypoxia and the various factors that contribute prenatal hypoxia with appropriate figures

Major concerns are given in detail 

Author Response

We thank the reviewer for the evaluation of the manuscript and feedback.

  1. The authors have added material to the section on nitrosative stress regarding the interaction between reactive nitrogen species (RNS) and reactive oxygen species (ROS), as suggested.
  2. The role of HIF in hypoxia development is multifaceted. It depends on the duration of the process and the specific member of the HIF family involved. HIFs exhibit cytoprotective properties under hypoxic conditions, stimulate reparative processes, and increase the concentration of free radical scavengers (heme oxygenase-1), VEGF, and angiopoietins. Under hypoxia, HIF-1 affects energy metabolism by regulating compensatory ATP synthesis shunts (such as the malate-aspartate shuttle), increases glutathione synthesis, and enhances cellular resistance to oxidative stress. It is known that HSP70 prolongs the "lifespan" of HIF-1. We have established that suppression of HIF-1 mRNA expression following intrauterine hypoxia occurs against the background of HSP70 deficiency. Numerous studies have demonstrated divergent changes in HIF-1 levels and its isoforms depending on the type and duration of hypoxia and the specific organ affected. Under conditions of enhanced nitrosative stress and elevated levels of cytotoxic NO products, along with ATP deficiency in tissues, a decrease in HIF is observed, associated with the activation of the ubiquitin-independent degradation pathway of oxidatively modified HIF-1α and suppression of its synthesis at the ATP-deficient stage. The authors believe it is more appropriate to refer to the pharmacological modulation of HIF. HIF inhibitors continue to be widely studied for clinical application, primarily in the treatment of cancer. However, HIF-α prolyl hydroxylase inhibitors are active compounds currently being considered for inducing physiological expression of erythropoietin, as well as promising cardioprotective and antihypoxic agents. The use of HIF-α prolyl hydroxylase inhibitors may be associated with risks, such as the activation of neovascularization (posing a potential oncogenic risk), increased thrombosis and cardiovascular complications, and heightened sensitivity of vascular pressor receptors.
  3. The authors have added a paragraph to the article dedicated to prenatal hypoxia and the various factors that contribute to its development.

Round 2

Reviewer 1 Report

The authors of the paper, Antioxidants-3600688, successfully addressed each and every one of the constructive observations I made about their work. Their work can now be better read and understood.

no tengo mas comentarios detallados sobre el trabajo.